# Understanding convolution on graphs via energies

**Francesco Di Giovanni**[*]                                   *fd405@cam.ac.uk*
*University of Cambridge*

**James Rowbottom**[*]                                         *jr908@cam.ac.uk*
*University of Cambridge*

**Benjamin P. Chamberlain**
*Charm Therapeutics*

**Thomas Markovich**
*Cash App*

**Michael M. Bronstein**
*University of Oxford*

**Reviewed on OpenReview:** *https://openreview.net/forum?id=v5ew3FPTgb*

## Abstract

Graph Neural Networks (GNNs) typically operate by message-passing, where the state of a node is updated based on the information received from its neighbours. Most message-passing models act as graph convolutions, where features are mixed by a shared, linear transformation before being propagated over the edges. On node-classification tasks, graph convolutions have been shown to suffer from two limitations: poor performance on heterophilic graphs, and over-smoothing. It is common belief that both phenomena occur because such models behave as low-pass filters, meaning that the Dirichlet energy of the features decreases along the layers incurring a smoothing effect that ultimately makes features no longer distinguishable. In this work, we rigorously prove that simple graph-convolutional models can actually enhance high frequencies and even lead to an asymptotic behaviour we refer to as *over-sharpening*, opposite to over-smoothing. We do so by showing that linear graph convolutions with symmetric weights minimize a multi-particle energy that generalizes the Dirichlet energy; in this setting, the weight matrices induce edge-wise attraction (repulsion) through their positive (negative) eigenvalues, thereby controlling whether the features are being smoothed or sharpened. We also extend the analysis to non-linear GNNs, and demonstrate that some existing time-continuous GNNs are instead always dominated by the low frequencies. Finally, we validate our theoretical findings through ablations and real-world experiments.

## 1 Introduction

GraphNeural Networks (GNNs) represent a popular class of neural networks operating on graphs (Sperduti, 1993; Goller & Kuchler, 1996; Gori et al., 2005; Scarselli et al., 2008; Bruna et al., 2014; Defferrard et al., 2016). Most GNNs follow the *message-passing* paradigm (Gilmer et al., 2017), where node embeddings are computed recursively after collecting information from the 1-hop neighbours. Typically, Message Passing Neural Networks (MPNNs) implement convolution on graphs, where messages are first acted upon by a linear transformation (referred to as *channel-mixing*), and are then propagated over the edges by a (normalized) adjacency matrix (Kipf & Welling, 2017; Hamilton et al., 2017; Xu et al., 2019; Bronstein et al., 2021).

While issues such as bounded expressive power (Xu et al., 2019; Morris et al., 2019) and over-squashing (Alon & Yahav, 2021; Topping et al., 2022; Di Giovanni et al., 2023) pertain to general MPNNs, graph-convolutional MPNNs have also been shown to suffer from additional problems more peculiar to node-classification tasks:

performance on *heterophilic* graphs – i.e. those where adjacent nodes often have different labels – and *over-smoothing* of features in the limit of many layers.

Several works have gathered evidence that graph convolutions seem to struggle on *some* heterophilic graphs (Pei et al., 2020; Zhu et al., 2020; Platonov et al., 2023). A common reasoning is that these MPNNs tend to enhance the low-frequency components of the features and discard the high-frequency ones, resulting in a smoothing effect which is detrimental on tasks where adjacent nodes have different labels (Nt & Maehara, 2019). Accordingly, a common fix for this problem is allowing signed-message passing to revert the smoothing effect (Bo et al., 2021; Yan et al., 2021). Nonetheless, the property that graph convolutions act as low-pass filters is actually only proven in very specialized scenarios, which leads to an important question:

> Q.1 Are simple graph convolutions *actually capable* of enhancing the high-frequency components of the features without explicitly modelling signed message-passing?

A positive answer to Q.1 might imply that, in the limit of many layers, simple graph convolutions can avoid over-smoothing. *Over-smoothing* occurs when node features become indistinguishable in the limit of many layers (Nt & Maehara, 2019; Oono & Suzuki, 2020). In fact, Cai & Wang (2020); Bodnar et al. (2022); Rusch et al. (2022) argued that over-smoothing is defined by the Dirichlet energy $\mathcal{E}^{\mathrm{Dir}}$ (Zhou & Schölkopf, 2005) decaying to zero (exponentially) as the depth increases. However, these theoretical works focus on a classical instance of GCN (Kipf & Welling, 2017) with assumptions on the non-linear activation and the singular values of the weight matrices, thereby leaving the following questions open:

> Q.2 Can over-smoothing be avoided by simple graph convolutions? If so, do graph convolutions admit asymptotic behaviours other than over-smoothing, in the limit of many layers?

**Contributions and outline.** Both the poor performance on heterophilic graphs and the issue of over-smoothing, can be characterized in terms of a *fixed energy functional* (the Dirichlet energy $\mathcal{E}^{\mathrm{Dir}}$) *decreasing along the features* $\mathbf{F}(t)$ *computed by graph-convolutional* MPNN*s*. Namely, if $\mathcal{E}^{\mathrm{Dir}}(\mathbf{F}(t))$ decreases w.r.t. the layer $t$, then the features become smoother and hence likely not useful for separating nodes on heterophilic graphs, ultimately incurring over-smoothing. But does actually $\mathcal{E}^{\mathrm{Dir}}(\mathbf{F}(t))$ decrease for all graph-convolutional models? In this work we show that for many graph convolutions, a more general, parametric energy $\mathcal{E}_\theta$, which recovers $\mathcal{E}^{\mathrm{Dir}}$ as a special case, decreases along the features as we increase the number of layers. Since this energy can also induce repulsion along the edges, we are able to provide affirmative answers to both Q.1 and Q.2 above, thereby showing that graph convolutions are not bound to act as low-pass filters and over-smooth in the limit. Namely, our contributions are:

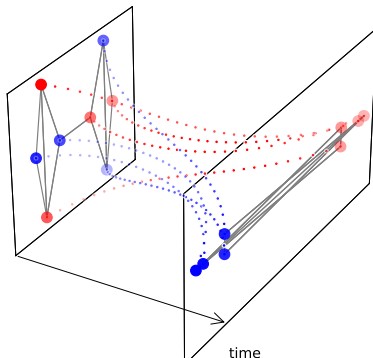

Figure 1: Actual gradient flow dynamics: repulsive forces separate heterophilic labels.

- In Section 4 we prove that a large class of linear graph convolutions are **gradient flows**, meaning that the features are evolved in the direction of steepest descent of some energy $\mathcal{E}_\theta$, as long as the weight (channel-mixing) matrices are symmetric. We study $\mathcal{E}_\theta$ and show that it provides a physical interpretation for graph convolutions as multi-particle dynamics where the weight matrices induce attraction (repulsion) on the edges, via their positive (negative) eigenvalues.

- In light of this, we provide an affirmative answer to Q.1, proving that the class of gradient-flow MPNNs can enhance the high-frequency components of the features (see Theorem 4.3 for the time-continuous case and Theorem 5.1 for the discrete setting). Our analysis suggests that gradient-flow MPNNs may be able to deal with heterophilic graphs, differently from other time-continuous MPNNs which instead can only induce smoothing of the features (Theorem 5.4), provided that a residual connection is available (Theorem 5.3).

- In Section 5 we also answer Q.2 positively by proving that in the limit of many layers, linear gradient-flow MPNNs admit two possible asymptotic behaviours. If the positive eigenvalues of the weight matrices are larger than the negative ones, then we incur over-smoothing; conversely, the phenomenon of *over-sharpening* occurs, where the features become dominated by the projection onto the eigenvector of the Laplacian associated with the highest frequency (Theorem 5.1). This result proves that asymptotic behaviours other than over-smoothing are possible, even for graph convolutions, depending on the interactions between the spectra of the graph Laplacian and of the weight matrices. To determine whether the underlying MPNN is smoothing/sharpening the features, we also introduce a measure based on the normalized Dirichlet energy, of independent interest.

- In Section 6 we extend our theoretical analysis to graph convolutions with non-linear activations, showing that despite not being gradient-flows, the same parametric energy $\mathcal{E}_\theta(\mathbf{F}(t))$ introduced above, is monotonically decreasing with respect to $t$, thereby preserving the interpretation of the weight matrices as edge-wise potentials inducing attraction (repulsion) via their eigenvalues (Theorem 6.1).

- Finally, in Section 7 we validate our theoretical findings through ablations and real-world experiments.

**Reproducibility.** Source code can be found at: https://github.com/JRowbottomGit/graff.

## 2 Graph convolutions and the Dirichlet energy

In this Section, we first review the message-passing paradigm, with emphasis on the class of convolutional-type MPNNs. We then introduce the *Dirichlet energy* on graphs, a known functional which can be used to study the performance of graph convolutions on heterophilic tasks and the over-smoothing phenomenon.

### 2.1 Preliminaries on graphs and the MPNN class

**Notations and conventions.** Consider a graph $\mathsf{G} = (\mathsf{V}, \mathsf{E})$, with $n := |\mathsf{V}|$ nodes and $\mathsf{E} \subset \mathsf{V} \times \mathsf{V}$ representing the edges. We assume that $\mathsf{G}$ is simple, *undirected*, and connected. Its adjacency matrix $\mathbf{A}$ is defined as $a_{ij} = 1$ if $(i, j) \in \mathsf{E}$ and zero otherwise. We let $\mathbf{D} = \mathrm{diag}(d_0, \ldots, d_{n-1})$ be the degree matrix. We typically use $\mathbf{A}$ to denote some (normalized) version of the adjacency matrix. We are interested in problems where the graph has node features $\{\mathbf{f}_i \in \mathbb{R}^d : i \in \mathsf{V}\}$ whose matrix representation is $\mathbf{F} \in \mathbb{R}^{n \times d}$. In particular, $\mathbf{f}_i \in \mathbb{R}^d$ is the $i$-th row (transposed) of $\mathbf{F}$, while $\mathbf{f}^r \in \mathbb{R}^n$ is its $r$-th column. Later, we also rely on the vectorization of the feature matrix $\mathrm{vec}(\mathbf{F}) \in \mathbb{R}^{nd}$, simply obtained by stacking all columns of $\mathbf{F}$.

**MPNNs and the 'convolutional flavour'.** In this work we investigate the dynamics of an MPNN in terms of its smoothing or sharpening effects on the features, which are properties particularly relevant in the context of heterophily and over-smoothing. Accordingly, we focus on node-level tasks, where each node $i \in \mathsf{V}$ has a label $y_i \in \{1, \ldots, K\}$ that we need to predict. Message Passing Neural Networks (MPNNs) (Gilmer et al., 2017) compute node embeddings $\mathbf{f}_i(t)$ at each node $i$ and layer $t$, using the following recursion:

$$\mathbf{f}_i(t+1) = \mathsf{UP}_t\left(\mathbf{f}_i(t), \mathsf{AGG}_t\left(\{\!\{\mathbf{f}_j(t) : (i, j) \in \mathsf{E}\}\!\}\right)\right),$$

where $\mathsf{AGG}_t$ is invariant to permutations. The over-smoothing phenomenon was studied for a class of MPNNs that we refer to as *graph convolutions* (Bronstein et al., 2021), which essentially consist of two operations: applying a shared linear transformation to the features (*'channel mixing'*) and propagating them along the edges (*'diffusion'*). Namely, we consider MPNNs whose layer update can be written in matricial form as:

$$\mathbf{F}(t+1) = \mathbf{F}(t) + \sigma\left(-\mathbf{F}(t)\mathbf{\Omega}_t + \mathbf{A}\mathbf{F}(t)\mathbf{W}_t - \mathbf{F}(0)\tilde{\mathbf{W}}_t\right), \tag{1}$$

where $\mathbf{\Omega}_t, \mathbf{W}_t$, and $\tilde{\mathbf{W}}_t$ are learnable matrices in $\mathbb{R}^{d \times d}$ performing channel mixing, while $\sigma$ is a pointwise nonlinearity; the message-passing matrix $\mathbf{A}$ instead, leads to the aggregation of features from adjacent nodes. We first note how this system of equations include common instances of the MPNN class with ***residual connections***. If $\mathbf{A} = \mathbf{D}^{-1/2}\mathbf{A}\mathbf{D}^{-1/2}$ and $\mathbf{\Omega}_t = \tilde{\mathbf{W}}_t = \mathbf{0}$, then we recover GCN (Kipf & Welling, 2017). The case of $\mathbf{A} = \mathbf{D}^{-1}\mathbf{A}$ and $\tilde{\mathbf{W}}_t = \mathbf{0}$ results in GraphSAGE (Hamilton et al., 2017), while by choosing $\mathbf{\Omega}_t = \mathbf{0}$ and $\mathbf{W}_t$ and $\tilde{\mathbf{W}}_t$ as convex combinations with the identity we recover GCNII (Chen et al., 2020). Finally, if $\mathbf{A} = \mathbf{A}$, $\mathbf{\Omega}_t = -(1 + \epsilon)\mathbf{W}_t$, and $\tilde{\mathbf{W}}_t = \mathbf{0}$, then (1) is a shallow variant of GIN (Xu et al., 2019).

## 2.2 The Dirichlet energy on graphs

To assess whether graph convolutions induce a smoothing or sharpening behaviour over the features, we monitor their Dirichlet energy. Consider a (feature) signal $\mathbf{F} : \mathsf{V} \to \mathbb{R}^d$, where we associate a vector $\mathbf{f}_i$ to each node $i \in \mathsf{V}$. The graph **Dirichlet energy** of $\mathbf{F}$ is defined by (Zhou & Schölkopf, 2005):

$$\mathcal{E}^{\mathrm{Dir}}(\mathbf{F}) := \frac{1}{2} \sum_{(i,j) \in \mathsf{E}} \|(\nabla \mathbf{F})_{ij}\|^2, \quad (\nabla \mathbf{F})_{ij} := \frac{\mathbf{f}_j}{\sqrt{d_j}} - \frac{\mathbf{f}_i}{\sqrt{d_i}},$$

where $\nabla \mathbf{F}$ is the *gradient* of $\mathbf{F}$ and is defined edge-wise. We see that $\mathcal{E}^{\mathrm{Dir}}$ measures the smoothness of $\mathbf{F}$, meaning whether $\mathbf{F}$ has similar values across adjacent nodes.

**The graph Laplacian.** The (normalized) graph *Laplacian* is the operator $\boldsymbol{\Delta} = \mathbf{I} - \mathbf{D}^{-1/2} \mathbf{A} \mathbf{D}^{-1/2}$. The Laplacian is symmetric and positive semidefinite, and its eigenvalues are referred to as (graph) *frequencies* and written in ascending order as $\lambda_0 \leq \lambda_1 \leq \ldots \leq \lambda_{n-1}$. It is known that $\lambda_0 = 0$ while $\lambda_{n-1} \leq 2$ (Chung & Graham, 1997). The low frequencies are associated with macroscopic (coarse) information in the graph, while the high frequencies generally describe microscopic (fine-grained) behaviour. In fact, if we let $\{\boldsymbol{\phi}_\ell\}$ be an orthonormal frame of eigenvectors for $\boldsymbol{\Delta}$ (i.e. $\boldsymbol{\Delta} \boldsymbol{\phi}_\ell = \lambda_\ell \boldsymbol{\phi}_\ell$), then we can rewrite the Dirichlet energy of the features $\mathbf{F}$ with vector representation $\mathrm{vec}(\mathbf{F}) \in \mathbb{R}^{nd}$ obtained by stacking all columns, as

$$\mathcal{E}^{\mathrm{Dir}}(\mathbf{F}) = \sum_{r=1}^{d} \sum_{\ell=0}^{n-1} \lambda_\ell (\boldsymbol{\phi}_\ell^\top \mathbf{f}^r)^2 = \mathrm{trace}(\mathbf{F}^\top \boldsymbol{\Delta} \mathbf{F}) = (\mathrm{vec}(\mathbf{F}))^\top (\mathbf{I}_d \otimes \boldsymbol{\Delta}) \mathrm{vec}(\mathbf{F}), \tag{2}$$

where $\otimes$ denotes the Kronecker product – some properties are reviewed in Appendix A.2 – and $\mathbf{I}_d$ is the identity matrix in $\mathbb{R}^{d \times d}$. Features are smooth if the Dirichlet energy is small, which occurs when for each channel $r$, we have large projections onto the eigenvectors $\boldsymbol{\phi}_\ell$ associated with lower frequencies.

## 3 Heterophily and over-smoothing: overview on related work

On a broad scale, our work is related to studying GNNs as filters (Defferrard et al., 2016; Bresson & Laurent, 2017; Hammond et al., 2019; Balcilar et al., 2020; He et al., 2021) and adopts techniques similar to Cai & Wang (2020). Our approach is also inspired by Haber & Ruthotto (2018); Chen et al. (2018); Biloš et al. (2021), where layers of an architecture are regarded as Euler discretizations of a time-continuous dynamical system. This direction has been studied in the context of GNNs in several flavours (Xhonneux et al., 2020; Zang & Wang, 2020; Chamberlain et al., 2021a; Eliasof et al., 2021; Chamberlain et al., 2021b; Bodnar et al., 2022; Rusch et al., 2022); we share this perspective in our work and expand the connection, by studying the energy associated with GNNs and think of features as 'particles' in some generalized position space.

**Understanding heterophily via the Dirichlet energy.** A node-classification task on a graph $\mathsf{G}$ is *heterophilic* when adjacent nodes often have different labels. In this case the label signal has high Dirichlet energy, meaning that enhancing the high-frequency components of the features could be useful to separate classes – the prototypical example being the eigenvector of $\boldsymbol{\Delta}$ with largest frequency $\lambda_{n-1}$, that can separate the clusters of a bipartite graph. While MPNNs as in (1) have shown strong empirical performance on node-classification tasks with low heterophily, it has been observed that simple convolutions on graphs could struggle in the case of high heterophily (Pei et al., 2020; Zhu et al., 2020). Such findings have spurred a plethora of methods that modify standard MPNNs to make them more effective on heterophilic graphs, by either ignoring or rearranging the graph structure, or by introducing some form of signed message-passing to 'reverse' the dynamics and magnify the high frequencies (Bo et al., 2021; Luan et al., 2021; Lim et al., 2021; Maurya et al., 2021; Zhang et al., 2021; Wang et al., 2022; Luan et al., 2022; Eliasof et al., 2023).

**Understanding over-smoothing via the Dirichlet energy.** Whether a given MPNN acts as a low-pass filter or not, is also at the heart of the *over-smoothing* phenomenon, regarded as the tendency of node features to approach the same value – up to degree rescaling – as the number of layers increase, independent of any input information (Li et al., 2018; Oono & Suzuki, 2020). Cai & Wang (2020) formalized over-smoothing as

the Dirichlet energy of the features $\mathcal{E}^{\mathrm{Dir}}(\mathbf{F}(t))$ decaying to zero as the number of layers $t$ diverge. A similar approach has also been studied in Zhou et al. (2021); Bodnar et al. (2022); Rusch et al. (2022) – for a review see Rusch et al. (2023). Although over-smoothing is now regarded as a general plague for MPNNs, it has in fact only been proven to occur for special instances of graph convolutional equations that (i) have nonlinear activation ReLU, (ii) have small weight matrices (as measured by their singular values), and (iii) have *no* residual connections.

**The general strategy.** Both the problem of graph convolutions struggling on heterophilic tasks and of over-smoothing, arise when a **fixed** energy functional, in this case the Dirichlet energy, decreases along the features computed at each layer. Namely, both these phenomena occur when, in general, $\frac{d}{dt}\mathcal{E}^{\mathrm{Dir}}(\mathbf{F}(t)) \leq 0$, with $\mathbf{F}(t)$ the features at layer $t$. To address the questions Q.1 and Q.2 in Section 1, *we study whether there exist general energy functionals promoting dynamics more expressive than simple smoothing of the features, that decrease along graph convolutions as in (1)*. Analyzing complicated dynamical systems through monotonicity properties of energy functionals is common in physics and geometry. Indeed, in this work we show that the same approach sheds light on the role of the channel-mixing weight matrices for graph-convolutions, and on the dynamics that MPNNs can generate beyond (over-)smoothing.

## 4 Gradient-flow MPNNs: understanding convolution on graphs via parametric energies

In this Section we study when graph-convolutions (1) admit an energy functional $\mathcal{E}_\theta$ such that $t \mapsto \mathcal{E}_\theta(\mathbf{F}(t))$ is decreasing as $t$ increases. This allows us to interpret the underlying MPNN-dynamics as forces that dissipate (or in fact, minimize) the energy $\mathcal{E}_\theta$. By analysing $\mathcal{E}_\theta$ we can then derive qualitative properties of the MPNN, and determine if the features $\mathbf{F}(t)$ incur over-smoothing or not when $t$ is large. Put differently, we investigate when MPNNs as in (1) are *gradient flows*, a special class of dynamical systems we briefly review next.

**What is a gradient flow?** Consider an $N$-dimensional dynamical system governed by the differential equation $\dot{\mathbf{F}}(t) = \mathcal{F}(\mathbf{F}(t))$ that evolves some input $\mathbf{F}(0)$ for time $t \geq 0$. We say that the evolution equation is a *gradient flow* if there exists $\mathcal{E} : \mathbb{R}^N \to \mathbb{R}$ such that $\mathcal{F}(\mathbf{F}(t)) = -\nabla\mathcal{E}(\mathbf{F}(t))$. In this case, since $\dot{\mathcal{E}}(\mathbf{F}(t)) = -\|\nabla\mathcal{E}(\mathbf{F}(t))\|^2$, the energy $\mathcal{E}$ *decreases* along the solution $\mathbf{F}(t)$. Gradient flows are valuable since the existence of $\mathcal{E}$ and the knowledge of its functional expression allow for a better understanding of the underlying dynamics. To motivate our analysis of graph convolutions, we first review examples of gradient flows on graphs that do not involve learnable weights – we discuss variational methods for image processing (You et al., 1996; Kimmel et al., 1997) in Appendix A.1 and Appendix B.5.

### 4.1 Gradient flows on graphs: the non-learnable case

**A prototypical gradient flow: heat equation.** Let $\mathbf{F} \in \mathbb{R}^{n \times d}$ be the feature matrix. The most common form of diffusion process on G is the *heat equation*, which consists of the system $\dot{\mathbf{f}}^r(t) = -\mathbf{\Delta}\mathbf{f}^r(t)$, where $\mathbf{f}^r \in \mathbb{R}^n$ is the $r$-th entry of the features for $1 \leq r \leq d$. The heat equation is an example of gradient flow: if we stack the columns of $\mathbf{F}$ into $\mathrm{vec}(\mathbf{F}) \in \mathbb{R}^{nd}$, we can rewrite the heat equation as

$$\mathrm{vec}(\dot{\mathbf{F}}(t)) = -\frac{1}{2}\nabla\mathcal{E}^{\mathrm{Dir}}(\mathrm{vec}(\mathbf{F}(t))), \tag{3}$$

where $\mathcal{E}^{\mathrm{Dir}} : \mathbb{R}^{nd} \to \mathbb{R}$ is the (graph) Dirichlet energy defined in (2). The evolution of $\mathbf{F}(t)$ by the heat equation (3) decreases the Dirichlet energy $\mathcal{E}^{\mathrm{Dir}}(\mathbf{F}(t))$ and is hence a smoothing process; in the limit, $\mathcal{E}^{\mathrm{Dir}}(\mathbf{F}(t)) \to 0$, which is attained by the projection of the initial state $\mathbf{F}(0)$ onto the null space of the Laplacian.

**Label propagation.** Given a graph G and labels $\{\mathbf{y}_i\}$ on $\mathsf{V}_1 \subset \mathsf{V}$, assume we want to predict the labels on $\mathsf{V}_2 \subset \mathsf{V}$. Zhou & Schölkopf (2005) introduced *label propagation* (LP) to solve this task. First, the input labels are extended by setting $\mathbf{y}_i(0) = \mathbf{0}$ for each $i \in \mathsf{V} \setminus \mathsf{V}_1$. The labels are then updated recursively according to the following rule, that is derived as gradient flow of an energy $\mathcal{E}^{\mathrm{LP}}$ we want to be minimized:

$$\dot{\mathbf{Y}}(t) = -\frac{1}{2}\nabla\mathcal{E}^{\mathrm{LP}}(\mathbf{Y}(t)), \quad \mathcal{E}^{\mathrm{LP}}(\mathbf{Y}) := \mathcal{E}^{\mathrm{Dir}}(\mathbf{Y}) + \mu\|\mathbf{Y} - \mathbf{Y}(0)\|^2. \tag{4}$$

The gradient flow ensures that the prediction is attained by the minimizer of $\mathcal{E}^{\mathrm{LP}}$, which both enforces smoothness via $\mathcal{E}^{\mathrm{Dir}}$ and penalizes deviations from the available labels (soft boundary conditions).

**Motivations.** Our goal amounts to extending the gradient flow formalism from the parameter-free case to a deep learning setting which includes features. We investigate when graph convolutions as in (1) admit (parametric) energy functionals that decrease along the features computed at each layer.

## 4.2 Gradient flows on graphs: the learnable case

In the spirit of Haber & Ruthotto (2018); Chen et al. (2018), we regard the family of MPNNs introduced in (1) as the Euler discretization of a continuous dynamical system. More specifically, we can introduce a step size $\tau \in (0, 1]$ and rewrite the class of MPNNs as

$$\mathbf{F}(t + \tau) - \mathbf{F}(t) = \tau \sigma \left( -\mathbf{F}(t)\mathbf{\Omega}_t + \mathbf{A}\mathbf{F}(t)\mathbf{W}_t - \mathbf{F}(0)\tilde{\mathbf{W}}_t \right),$$

which then becomes the discretization of the system of differential equations

$$\dot{\mathbf{F}}(t) = \sigma \left( -\mathbf{F}(t)\mathbf{\Omega}_t + \mathbf{A}\mathbf{F}(t)\mathbf{W}_t - \mathbf{F}(0)\tilde{\mathbf{W}}_t \right),$$

where $\cdot$ denotes the time derivative. Understanding the mechanism underlying this dynamical system might shed light on the associated, discrete family of MPNNs. By this parallelism, we study both the time-continuous and time-discrete (i.e. layers) settings, and start by focusing on the former for the rest of the section.

### 4.2.1 Which energies are being minimized along graph-convolutional models?

Assume that $\mathbf{A}$ in (1) is **symmetric** and let $\mathbf{F}(0)$ be the input features. We introduce a parametric function $\mathcal{E}_\theta : \mathbb{R}^{nd} \to \mathbb{R}$, parameterised by $d \times d$ weight matrices $\mathbf{\Omega}$ and $\mathbf{W}$ of the form:

$$\mathcal{E}_\theta(\mathbf{F}) = \underbrace{\sum_i \langle \mathbf{f}_i, \mathbf{\Omega}\mathbf{f}_i \rangle}_{\mathcal{E}_{\mathbf{\Omega}}^{\mathrm{ext}}} - \underbrace{\sum_{i,j} \mathsf{A}_{ij} \langle \mathbf{f}_i, \mathbf{W}\mathbf{f}_j \rangle}_{\mathcal{E}_{\mathbf{W}}^{\mathrm{pair}}} + \underbrace{\varphi^0(\mathbf{F}, \mathbf{F}(0))}_{\mathcal{E}_{\varphi^0}^{\mathrm{source}}}, \tag{5}$$

Note that we can recover the energies in Section 4.1. If $\mathbf{\Omega} = \mathbf{W} = \mathbf{I}_d$ and $\varphi^0 = 0$, then $\mathcal{E}_\theta = \mathcal{E}^{\mathrm{Dir}}$ as per (2), while if $\varphi^0$ is an $L_2$-penalty, then $\mathcal{E}_\theta = \mathcal{E}^{\mathrm{LP}}$ as per (4). We can also recover harmonic energies on graphs (see Appendix B.5). Importantly, if we choose $\varphi^0(\mathbf{F}, \mathbf{F}(0)) = 2 \sum_i \langle \mathbf{f}_i, \tilde{\mathbf{W}}\mathbf{f}_i(0) \rangle$, for $\tilde{\mathbf{W}} \in \mathbb{R}^{d \times d}$, then we can rewrite (see Appendix B.1)

$$\mathcal{E}_\theta(\mathbf{F}) = \langle \mathrm{vec}(\mathbf{F}), (\mathbf{\Omega} \otimes \mathbf{I}_n - \mathbf{W} \otimes \mathbf{A})\mathrm{vec}(\mathbf{F}) + 2(\tilde{\mathbf{W}} \otimes \mathbf{I}_n)\mathrm{vec}(\mathbf{F}(0)) \rangle. \tag{6}$$

The *gradient flow* of $\mathcal{E}_\theta$ can then be simply derived as (once we divide the gradient by 2):

$$\dot{\mathbf{F}}(t) = -\frac{1}{2}\nabla_{\mathbf{F}}\mathcal{E}_\theta(\mathbf{F}(t)) = -\mathbf{F}(t)\left(\frac{\mathbf{\Omega} + \mathbf{\Omega}^\top}{2}\right) + \mathbf{A}\mathbf{F}(t)\left(\frac{\mathbf{W} + \mathbf{W}^\top}{2}\right) - \mathbf{F}(0)\tilde{\mathbf{W}}. \tag{7}$$

Since $\mathbf{\Omega}, \mathbf{W}$ appear in (7) in a symmetrized way, without loss of generality we can assume $\mathbf{\Omega}$ and $\mathbf{W}$ to be *symmetric* $d \times d$ channel mixing matrices. Therefore, (7) simplifies as

$$\dot{\mathbf{F}}(t) = -\mathbf{F}(t)\mathbf{\Omega} + \mathbf{A}\mathbf{F}(t)\mathbf{W} - \mathbf{F}(0)\tilde{\mathbf{W}}. \tag{8}$$

**Proposition 4.1.** *Assume that* $\mathsf{G}$ *has a non-trivial edge. The linear, time-continuous* MPNN*s of the form*

$$\dot{\mathbf{F}}(t) = -\mathbf{F}(t)\mathbf{\Omega} + \mathbf{A}\mathbf{F}(t)\mathbf{W} - \mathbf{F}(0)\tilde{\mathbf{W}},$$

*are gradient flows of the energy in* (6) *if and only if the weight matrices* $\mathbf{\Omega}$ *and* $\mathbf{W}$ *are symmetric.*

Therefore, a large class of linear MPNNs evolve the features in the direction of steepest descent of an energy, provided that the weight matrices are symmetric. Note that despite reducing the degrees of freedom, the symmetry of the weight matrices does not diminish their power (Hu et al., 2019).

### 4.2.2 Attraction and repulsion: a physics-inspired framework

Thanks to the existence of an energy $\mathcal{E}_\theta$, we can provide a simple explanation for the dynamics induced by gradient-flow MPNNs as pairwise forces acting among adjacent features and generating attraction and repulsion depending on the eigenvalues of the weight matrices.

**Why gradient flows? A multi-particle point of view.** Below, we think of node features as particles in $\mathbb{R}^d$ with energy $\mathcal{E}_\theta$. In (5), the first term $\mathcal{E}_\Omega^{\mathrm{ext}}$ is *independent of the pairwise interactions* and hence represents an 'external' energy in the feature space. The second term $\mathcal{E}_{\mathbf{W}}^{\mathrm{pair}}$ instead accounts for *pairwise interactions* along edges via the symmetric matrix $\mathbf{W}$ and hence represents an energy associated with the graph structure. For simplicity, we set the source term $\varphi^0$ to zero and write $\mathbf{W} = \boldsymbol{\Theta}_+^\top \boldsymbol{\Theta}_+ - \boldsymbol{\Theta}_-^\top \boldsymbol{\Theta}_-$, by decomposing it into components with positive and negative eigenvalues. We can then rewrite $\mathcal{E}_\theta$ in (5) as

$$\mathcal{E}_\theta(\mathbf{F}) = \underbrace{\sum_i \langle \mathbf{f}_i, (\boldsymbol{\Omega} - \mathbf{W})\mathbf{f}_i \rangle}_{\textit{graph-independent}} + \underbrace{\frac{1}{2}\sum_{i,j} \|\boldsymbol{\Theta}_+(\nabla \mathbf{F})_{ij}\|^2}_{\textit{attraction}} - \underbrace{\frac{1}{2}\sum_{i,j} \|\boldsymbol{\Theta}_-(\nabla \mathbf{F})_{ij}\|^2}_{\textit{repulsion}}, \tag{9}$$

which we have derived in Appendix B. Consider now the gradient flow (8) where features $\mathbf{F}(t)$ are evolved in the direction of steepest descent of $\mathcal{E}_\theta$. Recall that the edge gradient $(\nabla \mathbf{F}(t))_{ij}$ is defined edge-wise and measures the difference between features $\mathbf{f}_i(t)$ and $\mathbf{f}_j(t)$ along $(i,j) \in \mathsf{E}$. We note that:

(i) The channel-mixing $\mathbf{W}$ encodes *attractive edge-wise interactions* via its positive eigenvalues since the gradient terms $\|\boldsymbol{\Theta}_+(\nabla \mathbf{F}(t))_{ij}\|$ are being minimized along (8), resulting in a smoothing effect where the edge-gradient is shrinking along the eigenvectors of $\mathbf{W}$ associated with positive eigenvalues;

(ii) The channel-mixing $\mathbf{W}$ encodes *repulsive edge-wise interactions* via its negative eigenvalues since the gradient terms $-\|\boldsymbol{\Theta}_-(\nabla \mathbf{F}(t))_{ij}\|$ are being minimized along (8), resulting in a sharpening effect where the edge-gradient is expanding along the eigenvectors of $\mathbf{W}$ associated with negative eigenvalues.

Next, we formalize the smoothing vs sharpening effects discussed in (i) and (ii) in a more formal way.

### 4.3 Low vs high frequency dominant dynamics: a new measure

Attractive forces reduce the edge gradients and are associated with smoothing effects which magnify low frequencies, while repulsive forces increase the edge gradients and hence afford a sharpening action enhancing the high frequencies. To determine which frequency is dominating the dynamics, we propose to monitor the *normalized* Dirichlet energy: $\mathcal{E}^{\mathrm{Dir}}(\mathbf{F}(t))/\|\mathbf{F}(t)\|^2$. This is the *Rayleigh quotient* of $\mathbf{I}_d \otimes \boldsymbol{\Delta}$ and so it satisfies $0 \leq \mathcal{E}^{\mathrm{Dir}}(\mathbf{F})/\|\mathbf{F}\|^2 \leq \lambda_{n-1}$ (see Appendix A.2). If this Rayleigh quotient is approaching its minimum, then the lowest frequency component is dominating, whereas if it is approaching its maximum, then the dynamics is dominated by the highest frequencies. This allows us to introduce the following characterization, of independent interest, to *measure the frequency-response* of spatial MPNNs:

**Definition 4.2.** Given a time-continuous (discrete) MPNN computing features $\mathbf{F}(t)$ at each time (layer) $t$, we say that the MPNN is *Low-Frequency-Dominant* (LFD) if $\mathcal{E}^{\mathrm{Dir}}(\mathbf{F}(t))/\|\mathbf{F}(t)\|^2 \to 0$ for $t \to \infty$. Conversely, we say that the MPNN is *High-Frequency-Dominant* (HFD) if $\mathcal{E}^{\mathrm{Dir}}(\mathbf{F}(t))/\|\mathbf{F}(t)\|^2 \to \lambda_{n-1}$ as $t \to \infty$.

Note that the LFD characterization differs from the notion of over-smoothing introduced in Cai & Wang (2020); Rusch et al. (2022), since it also accounts for the norm of the features. In fact, in Appendix B.2 we derive how our formulation also captures those cases where the Dirichlet energy is not converging to zero, yet the lowest frequency component is growing the fastest as time increases. Our notion of HFD-dynamics is also novel since previous works typically focused on showing when graph convolutions behave as low-pass filters.

Similarly to previous works, we now focus on a class of graph-convolutions inspired by GCN and show that even if we remove the terms $\boldsymbol{\Omega}$ and $\tilde{\mathbf{W}}$ from (1), graph-convolutions can enhance the high frequencies and ultimately lead to a HFD dynamics opposite to over-smoothing.

**Assumption.** We let $\mathbf{A} = \mathbf{D}^{-1/2}\mathbf{A}\mathbf{D}^{-1/2}$ and consider the simplified gradient flows with $\boldsymbol{\Omega} = \tilde{\mathbf{W}} = \mathbf{0}$.

**Theorem 4.3.** *Given a continuous* MPNN *of the form* $\dot{\mathbf{F}}(t) = \mathbf{A}\mathbf{F}(t)\mathbf{W}$, *let* $\mu_0 < \mu_1 \leq \ldots \leq \mu_{d-1}$ *be the eigenvalues of* $\mathbf{W}$. *If* $|\mu_0|(\lambda_{n-1} - 1) > \mu_{d-1}$, *then for almost every* $\mathbf{F}(0)$, *the* MPNN *is* HFD. *Conversely, if* $|\mu_0|(\lambda_{n-1} - 1) < \mu_{d-1}$, *then for almost every input* $\mathbf{F}(0)$, *the* MPNN *is* LFD.

We provide convergence rates in Theorem B.3 in the Appendix. In accordance with (9), Theorem 4.3 shows that the channel-mixing matrix $\mathbf{W}$ can generate both attractive and repulsive forces along edges, leading to either an LFD or a HFD dynamics. In fact, the MPNN is HFD when $\mathbf{W}$ has a negative eigenvalue $\mu_0$ sufficiently larger than the most positive one $\mu_{d-1}$ – and viceversa for the LFD case. Therefore, Theorem 4.3 provides affirmative answers to Q.1, Q.2 in Section 1 in the time-continuous case, by showing that a simple class of gradient flow convolutions on graphs can learn to enhance the high frequencies and lead to a HFD dynamics where the highest frequency components dominate in the limit of many layers.

**Remark.** We note that $\mathcal{E}_\theta$ can generally be negative and unbounded. With slight abuse of nomenclature, we call them 'energy' since the associated MPNNs follow the direction of steepest descent of such functional.

## 5 The interactions between the graph and channel-mixing spectra

In this Section we consider discretized gradient flows, so that we can extend the analysis to the case of layers of an architecture. We consider MPNNs that we derive by taking the Euler discretization of the gradient flow equations in Theorem 4.3; given a step size $\tau$ and $\mathbf{W} \in \mathbb{R}^{d \times d}$ *symmetric*, we have

$$\mathbf{F}(t + \tau) = \mathbf{F}(t) + \tau \mathbf{A}\mathbf{F}(t)\mathbf{W}, \quad \mathbf{F}(0) = \psi_{\mathrm{EN}}(\mathbf{F}_0), \tag{10}$$

where an *encoder* $\psi_{\mathrm{EN}} : \mathbb{R}^{n \times p} \to \mathbb{R}^{n \times d}$ processes input features $\mathbf{F}_0$ and the prediction $\psi_{\mathrm{DE}}(\mathbf{F}(T))$ is produced by a *decoder* $\psi_{\mathrm{DE}} : \mathbb{R}^{n \times d} \to \mathbb{R}^{n \times K}$. Here, $K$ is the number of label classes, $T = m\tau$ is the *integration time*, and $m$ is the number of *layers*. We note that (i) typically $\psi_{\mathrm{EN}}, \psi_{\mathrm{DE}}$ are MLPs, making the entire framework in (10) non-linear; (ii) since we have a residual connection, this is *not* equivalent to collapsing the dynamics into a single layer with aggregation matrix $\mathbf{A}^m$ as done in Wu et al. (2019) — see (31) in the Appendix.

### 5.1 Discrete gradient flows and spectral analysis

The gradient flow in (10) is a linear, residual GCN with nonlinear encoder and decoder operations. Once we vectorize the features $\mathbf{F}(t) \mapsto \mathrm{vec}(\mathbf{F}(t)) \in \mathbb{R}^{nd}$, we can rewrite the update as $\mathrm{vec}(\mathbf{F}(t + \tau)) = \mathrm{vec}(\mathbf{F}(t)) + \tau(\mathbf{W} \otimes \mathbf{A})\mathrm{vec}(\mathbf{F}(t))$ (see Appendix A.2 for details). In particular, if we pick bases $\{\psi_r\} \subset \mathbb{R}^d$ and $\{\phi_\ell\} \subset \mathbb{R}^n$ of orthonormal eigenvectors for $\mathbf{W}$ and $\mathbf{\Delta}$ respectively, *we can write the features after $m$ layers explicitly*:

$$\mathrm{vec}(\mathbf{F}(m\tau)) = \sum_{r=0}^{d-1}\sum_{\ell=0}^{n-1} (1 + \tau\mu_r(1 - \lambda_\ell))^m c_{r,\ell}(0)\psi_r \otimes \phi_\ell, \tag{11}$$

where $c_{r,\ell}(0) := \langle \mathrm{vec}(\mathbf{F}(0)), \psi_r \otimes \phi_\ell \rangle$ and $\{\mu_r\}$ are the eigenvalues of $\mathbf{W}$. We see that the interaction of the spectra $\{\mu_r\}$ and $\{\lambda_\ell\}$ is the 'driving' factor for the dynamics, with positive (negative) eigenvalues of $\mathbf{W}$ magnifying the frequencies $\lambda_\ell < 1$ ($> 1$ respectively). In fact, note that the projection of the features onto the kernel of $\mathbf{W}$ *stay invariant*. In the following we let $\mu_0 \leq \mu_1 \leq \ldots \leq \mu_{d-1}$ be in ascending order. Note that $\phi_{n-1}$ is the Laplacian eigenvector associated with largest frequency $\lambda_{n-1}$. We formulate the result below in terms of the following:

$$\frac{\mu_{d-1}}{\lambda_{n-1} - 1} < |\mu_0| < \frac{2}{\tau(2 - \lambda_{n-1})}. \tag{12}$$

Note that if (12) holds, then $\mu_0 < 0$ since $\lambda_{n-1} - 1 < 1$ whenever G is not bipartite. The first inequality means that the negative eigenvalues of $\mathbf{W}$ dominate the positive ones (once we factor in the graph spectrum contribution), while the second is a constraint on the step-size since if $\tau$ is too large, then we no longer approximate the gradient flow in (8). We restrict to the case where $\mu_0, \mu_{d-1}$ are simple eigenvalues, but extending the results to the degenerate case is straightforward.

**Theorem 5.1.** *Consider an* MPNN *with update rule* $\mathbf{F}(t+\tau) = \mathbf{F}(t) + \tau\mathbf{AF}(t)\mathbf{W}$, *with* $\mathbf{W}$ *symmetric. Given $m$ layers, if (12) holds, then there exists $\delta < 1$ s.t. for all $i \in \mathsf{V}$ we have:*

$$\mathbf{f}_i(m\tau) = (1 + \tau|\mu_0|(\lambda_{n-1}-1))^m \left(c_{0,n-1}(0)\,\boldsymbol{\phi}_{n-1}(i)\cdot\boldsymbol{\psi}_0 + \mathcal{O}\left(\delta^m\right)\right). \tag{13}$$

*Conversely, if $\mu_{d-1} > |\mu_0|(\lambda_{n-1}-1)$, then*

$$\mathbf{f}_i(m\tau) = (1 + \tau\mu_{d-1})^m \left(c_{d-1,0}(0)\sqrt{\frac{d_i}{2|\mathsf{E}|}}\cdot\boldsymbol{\psi}_{d-1} + \mathcal{O}\left(\delta^m\right)\right). \tag{14}$$

We first note that $\delta$ is reported in (32) in Appendix C.1. If (13) holds, then repulsive forces (high frequencies) dominate since for all $i \in \mathsf{V}$, we have $\mathbf{f}_i(m\tau) \sim \boldsymbol{\phi}_{n-1}(i)\cdot\boldsymbol{\psi}_0$, up to *lower order terms in the number of layers*. Thus as we increase the depth, any feature $\mathbf{f}_i(m\tau)$ becomes dominated by a multiple of $\boldsymbol{\psi}_0 \in \mathbb{R}^d$ that only changes based on the value of the Laplacian eigenvector $\boldsymbol{\phi}_{n-1}$ at node $i$. Conversely, if (14) holds, then $\mathbf{f}_i(m\tau) \sim \sqrt{d_i}\cdot\boldsymbol{\psi}_{d-1}$, meaning that the features become dominated by a multiple of $\boldsymbol{\psi}_{d-1}$ only depending on the degree of $i$ – which recovers the over-smoothing phenomenon (Oono & Suzuki, 2020).

**Corollary 5.2.** *If (13) holds, then the* MPNN *is* HFD *for almost every* $\mathbf{F}(0)$ *and* $\mathbf{F}(m\tau)/\|\mathbf{F}(m\tau)\|$ *converges to* $\mathbf{F}_\infty$ *s.t.* $\boldsymbol{\Delta}\mathbf{f}_\infty^r = \lambda_{n-1}\mathbf{f}_\infty^r$ *for each $r$. Conversely, if (14) holds, then the* MPNN *is* LFD *for almost every* $\mathbf{F}(0)$ *and* $\mathbf{F}(m\tau)/\|\mathbf{F}(m\tau)\|$ *converges to* $\mathbf{F}_\infty$ *s.t.* $\boldsymbol{\Delta}\mathbf{f}_\infty^r = \mathbf{0}$ *for each $r$.*

**Over-smoothing and over-sharpening.** Our analysis shows that (i) linear graph-convolutional equations can induce a sharpening effect (13) (which answers Q.1 in Section 1 affirmatively); (ii) graph convolutions can avoid over-smoothing through the negative eigenvalues of the channel-mixing by incurring the opposite behaviour of *over-sharpening* (i.e. HFD behaviour) in the limit of many layers, which provides an affirmative answer to Q.2. As per Corollary 5.2, over-sharpening entails that the features converge, up to rescaling, to the eigenvector of $\boldsymbol{\Delta}$ associated with the largest frequency. Similarly, Corollary 5.2 implies that LFD is a rigorous characterization of over-smoothing, given that features converge to a vector with zero Dirichlet energy in the limit of many layers. Accordingly, both over-smoothing and over-sharpening afford a loss of information, since the features converge to the Laplacian eigenvector with minimal or maximal Dirichlet energy, respectively. When the number of layers is small though, the features $\mathbf{F}(m\tau)$ also depend on the lower-order terms of the asymptotic expansion in Theorem 5.1; whether the dynamics is LFD or HFD will then affect if the lower or higher frequencies have a larger contribution to the prediction.

**The role of the residual connection.** The following result shows that the residual connection is crucial for the emergence of the over-sharpening (HFD) regime:

**Theorem 5.3.** *If* $\mathsf{G}$ *is not bipartite, and we remove the residual connection, i.e.* $\mathbf{F}(t+\tau) = \tau\mathbf{AF}(t)\mathbf{W}$, *with* $\mathbf{W}$ *symmetric, then the dynamics is* LFD *for almost every* $\mathbf{F}(0)$, *independent of the spectrum of* $\mathbf{W}$.

We see that the residual connection *enables the channel-mixing to steer the evolution* towards low or high frequencies depending on the task. If we drop the residual connection, $\mathbf{W}$ is less powerful and the only asymptotic regime is over-smoothing (LFD), independent of the spectrum of $\mathbf{W}$.

## 5.2 Frequency response and heterophily

**Negative eigenvalues flip the edge signs.** Let $\mathbf{W} = \boldsymbol{\Psi}\text{diag}(\boldsymbol{\mu})\boldsymbol{\Psi}^\top$ be the eigendecomposition of $\mathbf{W}$ yielding the Fourier coefficients $\mathbf{Z}(t) = \mathbf{F}(t)\boldsymbol{\Psi}$. We rewrite the discretized gradient flow $\mathbf{F}(t+\tau) = \mathbf{F}(t) + \tau\mathbf{AF}(t)\mathbf{W}$ in the Fourier domain of $\mathbf{W}$ as $\mathbf{Z}(t+\tau) = \mathbf{Z}(t) + \tau\mathbf{AZ}(t)\text{diag}(\boldsymbol{\mu})$ and note that along the eigenvectors of $\mathbf{W}$, if $\mu_r < 0$, then the dynamics is *equivalent* to flipping the sign of the edges. Therefore, negative edge-weight mechanisms as those proposed in Bo et al. (2021); Yan et al. (2021) to deal with heterophilic graphs, can in fact be simply achieved by a *residual* graph convolutional model where the channel-mixing matrix $\mathbf{W}$ has negative eigenvalues. We refer to (33) in the appendix for more details.

**Can other 'time-continuous' MPNNs be HFD?** A framework that cannot enhance the high frequencies may struggle on heterophilic datasets. While by Theorem 5.1 we know that gradient-flow MPNNs can be HFD,

we prove that some existing time-continuous MPNNs – CGNN (Xhonneux et al., 2020), GRAND (Chamberlain et al., 2021a), PDE – GCN$_D$ Eliasof et al. (2021) – are never HFD; in fact, in our experiments (Table 3) we validate that these models are more vulnerable to heterophily – for an explicit statement including convergence rates we refer to Theorem B.4.

**Theorem 5.4** (Informal). CGNN, GRAND *and* PDE − GCN$_D$ *induce smoothing and are never* HFD.

**Summary of the theoretical results.** We have analysed a class of linear MPNNs that represent discrete gradient flows of $\mathcal{E}_\theta$. This provides a 'multi-particle' interpretation for graph convolutions and sheds light onto the dynamics they generate. We have shown that the interaction between the eigenvectors and spectra of $\mathbf{W}$ and $\mathbf{\Delta}$ is what drives the dynamics. We have also proven how such interaction ultimately leads to two opposite asymptotic regimes referred to as over-smoothing (LFD) and over-sharpening (HFD). This has allowed us to draw connections with more recent frameworks, proving that simple convolutional models are not necessarily bound to induce smoothing among features, *provided that we have a residual connection*, and can indeed also magnify the high frequencies, a property that may be desirable on heterophilic tasks, ultimately providing affirmative answers to Q.1 and Q.2 in Section 1.

# 6 Extending the analysis to non-linear layers: the family of energy-dissipating MPNNs

As discussed above, analysing energies along MPNNs is a valuable approach for investigating their dynamics. Cai & Wang (2020); Bodnar et al. (2022) showed that, under some assumptions, $\mathcal{E}^{\mathrm{Dir}}$ is decreasing (exponentially) along some classes of graph convolutions, implying *over-smoothing* – see also Rusch et al. (2022). Consider the general family of (time-continuous) graph convolutions as in (1), with $\sigma$ a nonlinear activation:

$$\dot{\mathbf{F}}(t) = \sigma\left(-\mathbf{F}(t)\mathbf{\Omega} + \mathbf{A}\mathbf{F}(t)\mathbf{W} - \mathbf{F}(0)\tilde{\mathbf{W}}\right). \tag{15}$$

Although this is no longer a gradient flow (unless $\sigma$ is linear), we prove that if the weights are symmetric, then the energy $\mathcal{E}_\theta$ in (6) still decreases along (15).

**Theorem 6.1.** *Consider* $\sigma : \mathbb{R} \to \mathbb{R}$ *satisfying* $x \mapsto x\sigma(x) \geq 0$. *If* $\mathbf{F}$ *solves (15) with* $\mathbf{\Omega}, \mathbf{W}$ *being symmetric, then* $t \mapsto \mathcal{E}_\theta(\mathbf{F}(t))$ *is decreasing. If we discretize the system with step size* $\tau$ *and let* $c$ *denote the most positive eigenvalue of* $\mathbf{\Omega} \otimes \mathbf{I}_n - \mathbf{W} \otimes \mathbf{A}$ *– if no positive eigenvalues exists take* $c = 0$ *– then*

$$\mathcal{E}_\theta(\mathbf{F}(t+\tau)) - \mathcal{E}_\theta(\mathbf{F}(t)) \leq c\|\mathbf{F}(t+\tau) - \mathbf{F}(t)\|^2.$$

An important consequence of Theorem 6.1 is that for non-linear graph convolutions with symmetric weights, the physics interpretation is preserved since the same multi-particle energy $\mathcal{E}_\theta$ in (9) dissipates along the features. Note that in the time-discrete setting, the inequality can be interpreted as a Lipschitz regularity result. In general, in the nonlinear case we are no longer able to derive exact results in the limit of many layers, yet the channel-mixing $\mathbf{W}$ still induces attraction/repulsion along edges via its positive/negative eigenvalues (see Lemma D.1). Theorem 6.1 differs from Cai & Wang (2020); Bodnar et al. (2022) in two ways: (i) It asserts monotonicity of an energy $\mathcal{E}_\theta$ more general than $\mathcal{E}^{\mathrm{Dir}}$, since it is parametric and in fact also able to enhance the high frequencies; (ii) it holds for an infinite class of non-linear activations (beyond ReLU).

# 7 Experimental validation of the theoretical results

In this Section we validate the theoretical results through ablations and real-world experiments on node classification tasks. First, we study a linear, gradient-flow MPNN of the form:

$$\mathbf{F}(t+1) = \mathbf{F}(t) + \mathbf{A}\mathbf{F}(t)\mathbf{W}_S, \quad \mathbf{W}_S := (\mathbf{W} + \mathbf{W}^\top)/2, \tag{16}$$

where $\mathbf{W} \in \mathbb{R}^{d \times d}$ is a learnable weight matrix. Thanks to Proposition 4.1 we know that this is a discretized gradient flow that minimizes $\mathcal{E}_\theta$ in (6). In particular, from Theorem 5.1 we derive that the positive eigenvalues of $\mathbf{W}_S$ induce attraction along the edges, while the negative eigenvalues generate repulsion. We note that while the MPNN-update is linear, the map associating a label to each node based on its input feature, is actually non-linear, because the encoder and decoder are, typically, non-linear MLPs. Since this family of

| Dataset | Texas | Wisconsin | Cornell | Film | Squirrel | Chameleon | Citeseer | Pubmed | Cora |
|---|---|---|---|---|---|---|---|---|---|
| Homophily | **0.11** | **0.21** | **0.30** | **0.22** | **0.22** | **0.23** | **0.74** | **0.80** | **0.81** |
| #Nodes | 183 | 251 | 183 | 7,600 | 5,201 | 2,277 | 3,327 | 18,717 | 2,708 |
| #Edges | 295 | 466 | 280 | 26,752 | 198,493 | 31,421 | 4,676 | 44,327 | 5,278 |
| GCN | $60.81 \pm 4.4$ | $51.57 \pm 3.51$ | $59.19 \pm 4.75$ | $29.96 \pm 0.84$ | $42.77 \pm 2.38$ | $62.63 \pm 2.02$ | $75.9 \pm 1.4$ | $87.31 \pm 0.44$ | $86.0 \pm 1.0$ |
| SGCN$_{gf}$ | $80.54 \pm 4.95$ | $82.75 \pm 4.54$ | $74.32 \pm 6.97$ | $33.72 \pm 0.76$ | $51.02 \pm 1.68$ | $68.57 \pm 1.73$ | $76.4 \pm 1.72$ | $88.39 \pm 0.39$ | $87.12 \pm 0.61$ |
| GCN$_{gf}$ | $84.86 \pm 4.22$ | $84.12 \pm 2.97$ | $77.3 \pm 7.85$ | $35.13 \pm 0.61$ | $50.84 \pm 1.92$ | $68.27 \pm 1.45$ | $76.82 \pm 1.59$ | $88.49 \pm 0.42$ | $87.79 \pm 0.93$ |

Table 1: Comparison of GCN and the models in (16) and (17) over datasets of varying homophily.

graph convolutions amount to a gradient-flow, simplified (i.e. linear) GCN with a residual-connection, we adopt the notation SGCN$_{gf}$ (we recall that $\mathbf{A} = \mathbf{D}^{-1/2}\mathbf{A}\mathbf{D}^{-1/2}$).

By Theorem 6.1, if we 'activate' (16) as

$$\mathbf{F}(t+1) = \mathbf{F}(t) + \sigma\Big(\mathbf{A}\mathbf{F}(t)\mathbf{W}_S\Big), \quad \mathbf{W}_S := (\mathbf{W} + \mathbf{W}^\top)/2 \tag{17}$$

with $\sigma$ s.t. $x\sigma(x) \geq 0$, then $\mathcal{E}_\theta$ in (6) is decreasing, so that we can think of such equations as more general 'approximate' gradient flows. With slight abuse of notations, we refer to (17) as GCN$_{gf}$, since this is just a GCN-model with a residual connection and symmetric weights, which ensure the monotonicity of the energy.

**Real-world experiments: performance on node-classification tasks.** By Theorem 5.1 and Theorem 6.1, both (16) and (17) can induce repulsion along the edges through the negative eigenvalues of $\mathbf{W}_S$. Therefore, these models should be more robust to the heterophily of the graph when compared to the classical implementation of GCN – in fact, note that a linear GCN with symmetric weights is always LFD as per Theorem 5.3, and that non-linear GCN is bound to over-smooth if the singular values are sufficiently small (Oono & Suzuki, 2020; Cai & Wang, 2020). To validate this point, we run all three models on node classification tasks defined over graphs with varying homophily (Sen et al., 2008; Rozemberczki et al., 2021; Pei et al., 2020) (details in Appendix E). Training, validation and test splits are taken from Pei et al. (2020) for all datasets for comparison. The results are reported in Table 1. For all the datasets shown, we performed a simple, grid search over the space $m \in \{2, 4, 8\}$ – recall that $m$ is the depth – learning rate $\in \{0.001, 0.005\}$ and decay $\in \{0.0005, 0.005, 0.05\}$. We find that both (16) and (17) surpass the performance of GCN, often by a significant margin, across all heterophilic datasets. We emphasize that this is in support of our theoretical findings, since we proved that ***thanks to a residual connection***, graph-convolutional models enable the channel-mixing to also generate repulsion along the edges through its negative eigenvalues hence compensating for the underlying heterophily of the graphs. In particular, the results also show that imposing a symmetrization over the weights has no negative impact on performance.

**Remark.** *It is possible to significantly improve the results by considering additional $\mathbf{\Omega}$ and $\tilde{\mathbf{W}}$-terms as in (10) and* **tuning***, achieving strong results that often surpass more complicated benchmarks specifically designed for heterophilic tasks. Our results then further bring about questions about the quality of these tasks and whether graph-convolutional models really need to be augmented or replaced for heterophilic tasks. Since the main goal of the Section amounts to validating the theory, we report these results in Appendix E.*

**Spectral analysis.** To corroborate our analysis, we also investigate spectral properties of the weight matrix $\mathbf{W}_S$. We report statistics of the gradient-flow MPNN in (16) after training, over 4 datasets (two heterophilic ones, and two homophilic ones). We recall that $\mu_{d-1}$ and $\mu_0$ are the most positive and most negative eigenvalues of $\mathbf{W}_S$, respectively.

| Dataset | Squirrel | Chameleon | Pubmed | Cora |
|---|---|---|---|---|
| Homophily | 0.22 | 0.23 | 0.8 | 0.81 |
| $-\mu_{d-1}/\mu_0$ | 0.68 | 0.71 | 2.43 | 9.58 |
| $\frac{\mathcal{E}^{\mathrm{Dir}}(\mathbf{F}(m))/\|\mathbf{F}(m)\|^2}{\mathcal{E}^{\mathrm{Dir}}(\mathbf{F}(0))/\|\mathbf{F}(0)\|^2}$ | 0.83 | 0.65 | 0.35 | 0.19 |

Table 2: Spectral properties of the weights of SGCN$_{gf}$ and behaviour of the (normalized) Dirichlet energy.

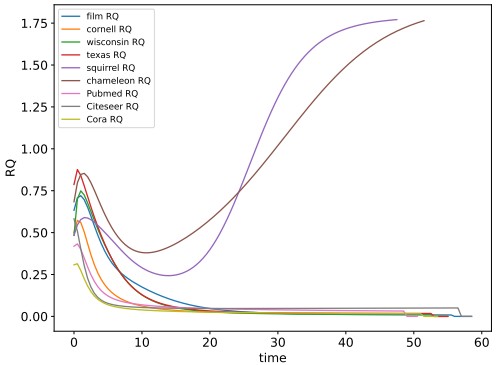 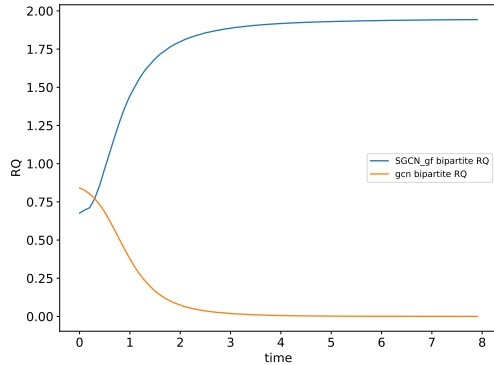

Figure 2: Rayleigh Quotient of $\mathbf{I}_d \otimes \mathbf{\Delta}$ (i.e. $\mathcal{E}^{\mathrm{Dir}}(\mathbf{F}(t))/\|\mathbf{F}(t)\|^2$) on real graphs for $\mathsf{SGCN}_{gf}$.

Figure 3: Rayleigh quotient of $\mathbf{I}_d \otimes \mathbf{\Delta}$ on a complete bipartite graph for $\mathsf{SGCN}_{gf}$ and $\mathsf{GCN}$.

In fact, by Theorem 5.1 we know that the ratio $-\mu_{d-1}/\mu_0$ indicates whether the underlying dynamics is dominated by the high frequencies (ratio smaller than 1), or by the low frequencies (ratio larger than one). We confirm this behaviour in Table 2, where we see that the quantity $-\mu_{d-1}/\mu_0$ is larger on the homophilic graphs and is instead smaller on the heterophilic graphs, where the model generally learns to enhance the high-frequency components more. We also monitor the profile of the normalized Dirichlet energy. We see that on homophilic graphs the low frequencies are indeed dominant since the normalized Dirichlet energy at the final layer is much lower than the one evaluated over the input features. This validates that our characterization of over-smoothing (LFD dynamics) and over-sharpening (HFD dynamics) is appropriate to study the dynamics of GNNs.

**Over-smoothing and over-sharpening.** We also validate our theoretical results that linear gradient-flows MPNNs as in (16) admit two, opposite, behaviours in the limit of many layers, depending on the eigenvalues of $\mathbf{W}_S$: (i) over-smoothing, i.e. LFD dynamics where the normalized Dirichlet energy approaches zero and (ii) over-sharpening, i.e. HFD dynamics where the normalized Dirichlet energy approaches the largest eigenvalue of the normalized Laplacian $\lambda_{n-1} \in [1, 2)$. Accordingly, we report the values of the normalized Dirichlet energy (i.e. the Rayleigh quotient of $\mathbf{I}_d \otimes \mathbf{\Delta}$, simply denoted by RQ) over several real-world datasets; here we are interested in validating that both over-smoothing and over-sharpening occur and that *no other* intermediate case is possible. Figure 2 confirms our theoretical analysis, showing that after training, the normalized Dirichlet energy approaches zero in 7 out of 9 cases (LFD dynamics, i.e. *over-smoothing*) whereas it converges to its maximum, depending on the graph, on the remaining two cases (HFD setting, i.e. *over-sharpening*), and that no other intermediate case arises. We point out that in this experiment we study the limiting behaviour of the underlying MPNNs and not the actual performance, which will of course degrade as we increase the number of layers for both over-smoothing and over-sharpening. To further support our claims, we also consider the synthetic case of a complete, bipartite graph with maximal heterophily (i.e. each node is connected to all other nodes with different label). In this extreme case, over-sharpening does actually provide the optimal classifier since the eigenvector of the graph Laplacian associated with largest frequency separates the two communities perfectly; in Figure 3 we see that for (16) the dynamics is HFD (i.e. over-sharpening), while a standard GCN over-smooths due to the lack of a residual connection as per Theorem 5.3.

## 8    Conclusions

**Summary and the messages of the work.** In this work we have shown that graph-convolutional models can enhance the high-frequency components of the features via the negative eigenvalues of the weight matrices, provided that there is a residual connection. In particular, we have studied a simple class of graph convolutions that represent the gradient flow of a parametric functional $\mathcal{E}_\theta$. The dynamics of gradient-flow MPNNs is fully characterized in terms of the interactions between the eigenvalues of the (symmetric) weight matrices and those of the graph Laplacian, ultimately leading to either over-smoothing or over-sharpening, in the limit of

many layers. We have also extended some of the conclusions to gradient flows activated by non-linear maps and have finally validated our analysis through ablation studies and real-world experiments.

Our work suggests that: (i) It is possible to shed light on the dynamics of GNNs once we identify an energy function that is decreasing along the features, thereby providing a general framework for analyzing existing models as well as building more principled ones; (ii) graph convolutions are not simple low-pass filters but can also learn to generate edge-wise repulsion once we have a residual connection; (iii) the role of heterophily as a major issue for GNNs and/or the validity of benchmarks commonly adopted in node classification, are brought into question, given that simple GCN models with residual connections are already on par, or better than GNN models designed specifically to deal with heterophily; (iv) the idea that over-smoothing is the dominant or only asymptotic behaviour for very deep MPNNs is questioned, considering that graph convolutions can also incur over-sharpening if the negative eigenvalues of the weight matrices are sufficiently large.

**Limitations and future work.** We limited our attention to a class of energy functionals whose gradient flows give rise to evolution equations of the graph-convolutional type. In future work, we plan to study other families of energies that generalize different GNN architectures and provide new models that are more 'physics'-inspired; we believe this to be a powerful paradigm to design new GNNs. To the best of our knowledge, our analysis is a first step into studying the interaction of the graph and 'channel-mixing' spectra; it will be interesting to explore more general dynamics (i.e., that are neither LFD nor HFD).

**Acknowledgements.** We thank the anonymous reviewers for valuable suggestions and feedback.

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

**Overview of the appendix**

To facilitate navigating the appendix, where we report several additional theoretical results, analysis of different cases along with further experiments and ablation studies, we provide the following detailed outline.

- In Appendix A.1 we review properties of the classical Dirichlet energy on manifolds that inspired traditional PDE variational methods for image processing whose extension to graphs and GNNs more specifically partly constitutes one of the main motivations of our work. We also review important elementary properties of the Kronecker product of matrices that are used throughout our proofs in Appendix A.2. We finally comment on the choice of the normalization (and symmetrization) of the graph Laplacian, briefly mentioning the impact of different choices.

- In Appendix B.1 we derive the energy decomposition reported in (9). In Appendix B.2 we derive additional rigorous results to justify our characterization of LFD and HFD dynamics in Definition 4.2 along with examples. We also formalize (and prove) more quantitatively Theorem 4.3 in Theorem B.3. In Appendix B.3 we report Theorem 5.4 including convergence rates and over-smoothing results. In Appendix B.4 we explore the special case of $\mathbf{\Omega} = \mathbf{W}$ which is equivalent to choosing $\mathbf{\Delta}$ rather than $\mathbf{A}$ as message-passing matrix, hence *providing new arguments as to why propagating messages using* $\mathbf{A}$ *rather than* $\mathbf{\Delta}$ *is 'more robust'*. Finally in Appendix B.5 we formally derive an analogy between the continuous energy used for manifolds (images) and a subset of the parametric energies in (5).

- In Appendix C we prove Theorem 5.1, Corollary 5.2, and Theorem 5.3.

- In Appendix D we prove Theorem 6.1 and an extra result confirming that even in the non-linear case the channel-mixing $\mathbf{W}$ still induces attraction and repulsion via its spectrum, thus magnifying the low or high frequencies respectively.

- In Appendix E we report additional details on the evaluation along with further real-world experiments and ablation studies.

**Additional notations and conventions used throughout the appendix.** Any graph $\mathsf{G}$ is taken to be *connected*. We order the eigenvalues of the graph Laplacian as $0 = \lambda_0 \leq \lambda_1 \leq \ldots \leq \lambda_{n-1} = \lambda_{n-1} \leq 2$ with associated orthonormal basis of eigenvectors $\{\phi_\ell\}_{\ell=0}^{n-1}$ so that in particular we have $\mathbf{\Delta}\phi_0 = \mathbf{0}$. Moreover, given a symmetric matrix $\mathbf{B}$, we generally denote the spectrum of $\mathbf{B}$ by $\mathrm{spec}(\mathbf{B})$ and if $\mathbf{B}$ is positive semi-definite (written as $\mathbf{B} \succeq 0$), then $\mathrm{gap}(\mathbf{B})$ denotes the *positive, smallest eigenvalue* of $\mathbf{B}$. Finally, if we write $\mathbf{F}(t)/\|\mathbf{F}(t)\|$ we always take the norm to be the Frobenius one and tacitly assume that the dynamics is s.t. the solution is not zero. Without losing generality, below we always take $\mathbf{A} = \mathbf{D}^{-1/2}\mathbf{A}\mathbf{D}^{-1/2}$.

## A  Motivations and important preliminaries

### A.1  Discussion on continuous Dirichlet energy and harmonic maps

**Starting point: a geometric parallelism.** To motivate a gradient-flow approach for GNNs, we start from the continuous case. Consider a smooth map $f : \mathbb{R}^n \to (\mathbb{R}^d, h)$ with $h$ a constant metric represented by $\mathbf{H} \succeq 0$. The *Dirichlet energy* of $f$ is defined by

$$\mathcal{E}(f, h) = \frac{1}{2} \int_{\mathbb{R}^n} \|\nabla f\|_h^2 \, dx = \frac{1}{2} \sum_{q,r=1}^{d} \sum_{j=1}^{n} \int_{\mathbb{R}^n} h_{qr} \partial_j f^q \partial_j f^r(x) dx \tag{18}$$

and measures the 'smoothness' of $f$. A natural approach to find minimizers of $\mathcal{E}$ - called *harmonic maps* - was introduced in Eells & Sampson (1964) and consists in studying the **gradient flow** of $\mathcal{E}$, wherein a given map $f(0) = f_0$ is evolved according to $\dot{f}(t) = -\nabla_f \mathcal{E}(f(t))$. These type of evolution equations have historically been the core of *variational* and *PDE-based image processing*; in particular, gradient flows of the Dirichlet energy (Kimmel et al., 1997), were shown to recover the Perona-Malik nonlinear diffusion Perona & Malik (1990).

### A.2 Review of Kronecker product and properties of Laplacian kernel

**Kronecker product.** In this subsection we summarize a few relevant notions pertaining to the Kronecker product of matrices, that are going to be applied throughout our spectral analysis of gradient flow equations for GNNs in both the continuous and discrete time setting.

Given a matricial equation of the form

$$\mathbf{Y} = \mathbf{AXB},$$

we can vectorize $\mathbf{X}$ and $\mathbf{Y}$ by stacking columns into $\text{vec}(\mathbf{X})$ and $\text{vec}(\mathbf{Y})$ respectively, and rewrite the previous system as

$$\text{vec}(\mathbf{Y}) = \left(\mathbf{B}^\top \otimes \mathbf{A}\right)\text{vec}(\mathbf{X}). \tag{19}$$

If $\mathbf{A}$ and $\mathbf{B}$ are symmetric with spectra $\text{spec}(\mathbf{A})$ and $\text{spec}(\mathbf{B})$ respectively, then the spectrum of $\mathbf{B} \otimes \mathbf{A}$ is given by $\text{spec}(\mathbf{A}) \cdot \text{spec}(\mathbf{B})$. Namely, if $\mathbf{Ax} = \lambda^\mathbf{A}\mathbf{x}$ and $\mathbf{By} = \lambda^\mathbf{B}\mathbf{y}$, for $\mathbf{x}$ and $\mathbf{y}$ non-zero vectors, then $\lambda^\mathbf{B}\lambda^\mathbf{A}$ is an eigenvalue of $\mathbf{B} \otimes \mathbf{A}$ with eigenvector $\mathbf{y} \otimes \mathbf{x}$:

$$(\mathbf{B} \otimes \mathbf{A})\,\mathbf{y} \otimes \mathbf{x} = (\lambda^\mathbf{B}\lambda^\mathbf{A})\mathbf{y} \otimes \mathbf{x}. \tag{20}$$

One can also define the *Kronecker sum* of matrices $\mathbf{A} \in \mathbb{R}^{n \times n}$ and $\mathbf{B} \in \mathbb{R}^{d \times d}$ as

$$\mathbf{A} \oplus \mathbf{B} := \mathbf{A} \otimes \mathbf{I}_d + \mathbf{I}_n \otimes \mathbf{B}, \tag{21}$$

with spectrum $\text{spec}(\mathbf{A} \oplus \mathbf{B}) = \{\lambda^\mathbf{A} + \lambda^\mathbf{B} : \lambda^\mathbf{A} \in \text{spec}(\mathbf{A}),\ \lambda^\mathbf{B} \in \text{spec}(\mathbf{B})\}$.

**Additional details on $\mathcal{E}^{\text{Dir}}$ and the choice of Laplacian.** We recall that the classical graph Dirichlet energy $\mathcal{E}^{\text{Dir}}$ is defined by

$$\mathcal{E}^{\text{Dir}}(\mathbf{F}) = \text{trace}\left(\mathbf{F}^\top \boldsymbol{\Delta}\mathbf{F}\right).$$

We can use the Kronecker product to rewrite the Dirichlet energy as

$$\mathcal{E}^{\text{Dir}}(\mathbf{F}) = \text{vec}(\mathbf{F})^\top(\mathbf{I}_d \otimes \boldsymbol{\Delta})\text{vec}(\mathbf{F}), \tag{22}$$

from which we immediately derive that $\frac{1}{2}\nabla_{\text{vec}(\mathbf{F})}\mathcal{E}^{\text{Dir}}(\mathbf{F}) = (\mathbf{I}_d \otimes \boldsymbol{\Delta})\text{vec}(\mathbf{F})$ – since $\boldsymbol{\Delta}$ is *symmetric* – and hence recover the gradient flow in (3) leading to the graph heat equation across each channel.

Before we further comment on the characterizations of LFD and HFD dynamics, we review the main choices of graph Laplacian and the associated harmonic signals (i.e. how we can characterize the kernel spaces of the given Laplacian operator). Recall that throughout the appendix we always assume that the underlying graph G is *connected*. The symmetrically normalized Laplacian $\boldsymbol{\Delta} = \mathbf{I} - \mathbf{A}$ is symmetric, positive semi-definite with harmonic space of the form (Chung & Graham, 1997)

$$\ker(\boldsymbol{\Delta}) := \text{span}(\mathbf{D}^{\frac{1}{2}}\mathbf{1}_n : \mathbf{1}_n = (1,\ldots,1)^\top). \tag{23}$$

Therefore, if the node features $\mathbf{F}(t)$ satisfy $\dot{\mathbf{F}}(t) = \text{GNN}_\theta(\mathbf{F}(t), t)$ with initial condition $\mathbf{F}(0)$, and we assume that over-smoothing occurs, meaning that $\boldsymbol{\Delta}\mathbf{f}^r(t) \to \mathbf{0}$ for $t \to \infty$ for each column $1 \leq r \leq d$, then the only information persisting in the asymptotic regime is the degree and any dependence on the input features is lost, as studied in Oono & Suzuki (2020); Cai & Wang (2020). A slightly different behaviour occurs if instead of $\boldsymbol{\Delta}$, we consider the unnormalized Laplacian $\mathbf{L} = \mathbf{D} - \mathbf{A}$ with kernel given by $\text{span}(\mathbf{1}_n)$, meaning that if $\mathbf{Lf}^r(t) \to \mathbf{0}$ as $t \to \infty$ for each $1 \leq r \leq d$, then any node would be embedded into a single point, hence making any separation task impossible. The same consequence applies to the random walk Laplacian $\boldsymbol{\Delta}_{\text{RW}} = \mathbf{I} - \mathbf{D}^{-1}\mathbf{A}$. In particular, we note that generally a row-stochastic matrix is not symmetric – if it was, then this would in fact be doubly-stochastic – and the same applies to the random-walk Laplacian (a special exception is given by the class of *regular* graphs). In fact, in general any dynamical system governed by $\boldsymbol{\Delta}_{\text{RW}}$ (or simply $\mathbf{D}^{-1}\mathbf{A}$) is not the gradient flow of an energy due to the lack of symmetry, as further confirmed below in (24).

## B  Proofs and additional details of Section 4

### B.1  Attraction vs repulsion: a physics-inspired framework

First, we show that we can rewrite the energy in (5) using the Kronecker product formalism. It suffices to focus on the term $\mathbf{\Omega}$-term since the other derivations are identical. We have

$$\sum_i \langle \mathbf{f}_i . \mathbf{\Omega} \mathbf{f}_i \rangle = \sum_i f_i^\alpha \Omega_{\alpha\beta} f_i^\beta.$$

On the other hand, since $(\mathbf{\Omega} \otimes \mathbf{I}) \mathrm{vec}(\mathbf{F}) = \mathbf{I} \mathbf{F} \mathbf{\Omega}^\top$, we also find

$$\langle \mathrm{vec}(\mathbf{F}), (\mathbf{\Omega} \otimes \mathbf{I}) \mathrm{vec}(\mathbf{F}) = f_i^\alpha (\mathbf{I} \mathbf{F} \mathbf{\Omega}^\top)_i^\alpha = f_i^\alpha \Omega_{\alpha\beta} f_i^\beta,$$

which shows the equivalence claimed.

*Proof of Proposition 4.1.* We first note that the system in (8) can be written using the Kronecker product as

$$\mathrm{vec}(\dot{\mathbf{F}}(t)) = -(\mathbf{\Omega} \otimes \mathbf{I}_n) \mathrm{vec}(\mathbf{F}(t)) + (\mathbf{W} \otimes \mathbf{A}) \mathrm{vec}(\mathbf{F}(t)) - (\tilde{\mathbf{W}} \otimes \mathbf{I}_n) \mathrm{vec}(\mathbf{F}(0)).$$

If this is the gradient flow of $\mathbf{F} \mapsto \mathcal{E}_\theta(\mathbf{F})$, then we would have

$$\nabla^2_{\mathrm{vec}(\mathbf{F})} \mathcal{E}_\theta(\mathbf{F}) = \mathbf{\Omega} \otimes \mathbf{I}_n - \mathbf{W} \otimes \mathbf{A}, \tag{24}$$

which must be symmetric due to the Hessian of a function being symmetric. The latter means

$$(\mathbf{\Omega}^\top - \mathbf{\Omega}) \otimes \mathbf{I}_n = (\mathbf{W}^\top - \mathbf{W}) \otimes \mathbf{A}.$$

Therefore, for each components $\alpha, \beta$ we must have

$$(\omega_{\alpha\beta} - \omega_{\beta\alpha}) \mathbf{I} = (w_{\alpha\beta} - w_{\beta\alpha}) \mathbf{A}.$$

If $\mathbf{A}$ is non-diagonal, which happens as soon as there is a non-trivial edge in $\mathsf{G}$, the equations are satisfied only if both $\mathbf{\Omega}$ and $\mathbf{W}$ are *symmetric*. This shows that (8) *is the gradient flow of $\mathcal{E}_\theta$ if and only if $\mathbf{\Omega}$ and $\mathbf{W}$ are symmetric.*

$\square$

We now rely on the spectral decomposition of $\mathbf{W}$ to rewrite $\mathcal{E}_\theta$ explicitly in terms of attractive and repulsive interactions. If we have a spectral decomposition $\mathbf{W} = \mathbf{\Psi} \mathrm{diag}(\boldsymbol{\mu})(\mathbf{\Psi})^\top$, we can separate the positive eigenvalues from the negative ones and write

$$\mathbf{W} = \mathbf{\Psi} \mathrm{diag}(\boldsymbol{\mu}_+) \mathbf{\Psi}^\top + \mathbf{\Psi} \mathrm{diag}(\boldsymbol{\mu}_-) \mathbf{\Psi}^\top := \mathbf{W}_+ - \mathbf{W}_-.$$

Since $\mathbf{W}_+ \succeq 0, \mathbf{W}_- \succeq 0$, we can use the Choleski decomposition to write $\mathbf{W}_+ = \mathbf{\Theta}_+^\top \mathbf{\Theta}_+$ and $\mathbf{W}_- = \mathbf{\Theta}_-^\top \mathbf{\Theta}_-$ with $\mathbf{\Theta}_+, \mathbf{\Theta}_- \in \mathbb{R}^{d \times d}$. Equation (9) then simply follows by direct computation: namely, if we let $(\mathbf{D}^{-1/2} \mathbf{A} \mathbf{D}^{-1/2})_{ij} = \bar{a}_{ij}$, we derive

$$\begin{aligned}
\mathcal{E}_\theta(\mathbf{F}) &= \sum_i \langle \mathbf{f}_i, \mathbf{\Omega} \mathbf{f}_i \rangle - \sum_{i,j} \bar{a}_{ij} \langle \mathbf{f}_i, \mathbf{W} \mathbf{f}_j \rangle \\
&= \sum_i \langle \mathbf{f}_i, (\mathbf{\Omega} - \mathbf{W}) \mathbf{f}_i \rangle + \sum_i \langle \mathbf{f}_i, \mathbf{W} \mathbf{f}_i \rangle - \sum_{i,j} \bar{a}_{ij} \langle \mathbf{\Theta}_+ \mathbf{f}_i, \mathbf{\Theta}_+ \mathbf{f}_j \rangle + \sum_{i,j} \bar{a}_{ij} \langle \mathbf{\Theta}_- \mathbf{f}_i, \mathbf{\Theta}_- \mathbf{f}_j \rangle \\
&= \sum_i \langle \mathbf{f}_i, (\mathbf{\Omega} - \mathbf{W}) \mathbf{f}_i \rangle + \frac{1}{2} \sum_{i,j} \|\mathbf{\Theta}_+ (\nabla \mathbf{F})_{ij}\|^2 - \frac{1}{2} \sum_{i,j} \|\mathbf{\Theta}_- (\nabla \mathbf{F})_{ij}\|^2,
\end{aligned}$$

where we have used that $\sum_{i,j} \frac{1}{d_i} \|\mathbf{\Theta}_+ \mathbf{f}_i\|^2 = \sum_i \|\mathbf{\Theta}_+ \mathbf{f}_i\|^2$.

## B.2 Additional details on LFD and HFD characterizations

In this subsection we provide further details and justifications for Definition 4.2. We first prove the following simple properties.

**Lemma B.1.** *Assume we have a (continuous) process $t \mapsto \mathbf{F}(t) \in \mathbb{R}^{n \times d}$, for $t \geq 0$. The following equivalent characterizations hold:*

(i) *$\mathcal{E}^{\mathrm{Dir}}(\mathbf{F}(t)) \to 0$ for $t \to \infty$ if and only if $\mathbf{\Delta f}^r(t) \to \mathbf{0}$, for $1 \leq r \leq d$.*

(ii) *$\mathcal{E}^{\mathrm{Dir}}(\mathbf{F}(t)/\|\mathbf{F}(t)\|) \to \lambda_{n-1}$ for $t \to \infty$ if and only if for any sequence $t_j \to \infty$ there exist a subsequence $t_{j_k} \to \infty$ and a unit limit $\mathbf{F}_\infty$ – depending on the subsequence – such that $\mathbf{\Delta f}_\infty^r = \lambda_{n-1}\mathbf{f}_\infty^r$, for $1 \leq r \leq d$.*

*Proof.* (i) Given $\mathbf{F}(t) \in \mathbb{R}^{n \times d}$, we can vectorize it and decompose it in the orthonormal basis $\{\mathbf{e}_r \otimes \boldsymbol{\phi}_\ell : 1 \leq r \leq d, \ 0 \leq \ell \leq n-1\}$, with $\{\mathbf{e}_r\}_{r=1}^d$ canonical basis in $\mathbb{R}^d$, and write

$$\mathrm{vec}(\mathbf{F}(t)) = \sum_{r,\ell} c_{r,\ell}(t)\mathbf{e}_r \otimes \boldsymbol{\phi}_\ell, \quad c_{r,\ell}(t) := \langle \mathrm{vec}(\mathbf{F}(t)), \mathbf{e}_r \otimes \boldsymbol{\phi}_\ell \rangle.$$

We can then use (22) to compute the Dirichlet energy as

$$\mathcal{E}^{\mathrm{Dir}}(\mathbf{F}(t)) = \sum_{r=1}^d \sum_{\ell=0}^{n-1} c_{r,\ell}^2(t)\lambda_\ell \equiv \sum_{r=1}^d \sum_{\ell=1}^{n-1} c_{r,\ell}^2(t)\lambda_\ell \geq \mathrm{gap}(\mathbf{\Delta}) \sum_{r=1}^d \sum_{\ell=1}^{n-1} c_{r,\ell}^2(t),$$

where we have used that $\lambda_0 = 0$ and that $\mathrm{gap}(\mathbf{\Delta}) = \lambda_1 \leq \lambda_\ell$ for all $\ell \geq 1$. Therefore

$$\mathcal{E}^{\mathrm{Dir}}(\mathbf{F}(t)) \to 0 \iff \sum_{r=1}^d \sum_{\ell=1}^{n-1} c_{r,\ell}^2(t) \to 0, \quad t \to \infty,$$

which occurs if and only if

$$(\mathbf{I}_d \otimes \mathbf{\Delta})\mathrm{vec}(\mathbf{F}(t)) = \sum_{r=1}^d \sum_{\ell=1}^{n-1} c_{r,\ell}(t)\lambda_\ell\mathbf{e}_r \otimes \boldsymbol{\phi}_\ell \to 0.$$

(ii) The argument here is similar. Indeed we can write $\mathbf{Q}(t) = \mathbf{F}(t)/\|\mathbf{F}(t)\|$ with $\mathbf{Q}(t)$ a unit-norm signal. Namely, we can vectorize it and write

$$\mathrm{vec}(\mathbf{Q}(t)) = \sum_{r,\ell} q_{r,\ell}(t)\mathbf{e}_r \otimes \boldsymbol{\phi}_\ell, \quad \sum_{r,\ell} q_{r,\ell}^2(t) = 1.$$

Then $\mathcal{E}^{\mathrm{Dir}}(\mathbf{Q}(t)) \to \lambda_{n-1}$ if and only if

$$\sum_{r,\ell} q_{r,\ell}^2(t)\lambda_\ell \to \lambda_{n-1}, \quad t \to \infty,$$

which holds if and only if

$$\sum_r q_{r,\lambda_{n-1}}^2(t) \to 1$$

$$q_{r,\ell}^2(t) \to 0, \quad \ell : \lambda_\ell < \lambda_{n-1}, \tag{25}$$

given the unit norm constraint. This is equivalent to the Rayleigh quotient of $\mathbf{I}_d \otimes \mathbf{\Delta}$ converging to its maximal value $\lambda_{n-1}$. When this occurs, for any sequence $t_j \to \infty$ we have that $q_{r,\ell}^2(t_j) \leq 1$, meaning that we can extract a converging subsequence that due to (25) will converge to a unit eigenvector $\mathbf{F}_\infty$ of $\mathbf{I}_d \otimes \mathbf{\Delta}$ satisfying $(\mathbf{I}_d \otimes \mathbf{\Delta})\mathbf{F}_\infty = \lambda_{n-1}\mathbf{Q}_\infty$. Conversely assume for a contradiction that there exists a sequence

$t_j \to \infty$ such that $\mathcal{E}^{\mathrm{Dir}}(\mathbf{F}(t_j)/\|\mathbf{F}(t_j)\|) < \lambda_{n-1} - \epsilon$, for some $\epsilon > 0$. Then (25) fails to be satisfied along the sequence, meaning that no subsequence converges to a unit norm eigenvector $\mathbf{F}_\infty$ of $\mathbf{I}_d \otimes \mathbf{\Delta}$ with associated eigenvalue $\lambda_{n-1}$ which is a contradiction to our assumption.

$\square$

Before we address the formulation of low(high)-frequency-dominated dynamics, we solve explicitly the system $\dot{\mathbf{F}}(t) = \mathbf{A}\mathbf{F}(t)$ in $\mathbb{R}^{n \times d}$, with some initial condition $\mathbf{F}(0)$. We can vectorize the equation and solve $\dot{\mathrm{vec}}(\mathbf{F}(t)) = (\mathbf{I}_d \otimes \mathbf{A})\mathrm{vec}(\mathbf{F}(t))$, meaning that

$$\mathrm{vec}(\mathbf{F}(t)) = \sum_{r=1}^{d} \sum_{\ell=0}^{n-1} e^{(1-\lambda_\ell)t} c_{r,\ell}(0) \mathbf{e}_r \otimes \boldsymbol{\phi}_\ell, \qquad c_{r,\ell}(0) := \langle \mathrm{vec}(\mathbf{F}(0)), \mathbf{e}_r \otimes \boldsymbol{\phi}_\ell \rangle.$$

Consider any initial condition $\mathbf{F}(0)$ such that

$$\sum_{r=1}^{d} |c_{r,0}| = \sum_{r=1}^{d} |\langle \mathrm{vec}(\mathbf{F}(0)), \mathbf{e}_r \otimes \boldsymbol{\phi}_0 \rangle| > 0,$$

which is satisfied for each $\mathrm{vec}(\mathbf{F}(0)) \in \mathbb{R}^{nd} \setminus \mathcal{U}^\perp$, where $\mathcal{U}^\perp$ is the orthogonal complement of $\mathbb{R}^d \otimes \mathrm{span}(\boldsymbol{\phi}_0)$. Since $\mathcal{U}^\perp$ is a lower-dimensional subspace, its complement is dense. Accordingly for a.e. $\mathbf{F}(0)$, we find that the solution satisfies

$$\|\mathrm{vec}(\mathbf{F}(t))\|^2 = e^{2t} \left( \sum_{r=1}^{d} c_{r,0}^2 + \mathcal{O}(e^{-2\mathrm{gap}(\mathbf{\Delta})t}) \right) = e^{2t} \left( \|P^\perp_{\ker(\mathbf{\Delta})} \mathrm{vec}(\mathbf{F}(0))\|^2 + \mathcal{O}(e^{-2\mathrm{gap}(\mathbf{\Delta})t}) \right),$$

with $P^\perp_{\ker(\mathbf{\Delta})}$ the projection onto $\mathbb{R}^d \otimes \ker(\mathbf{\Delta})$. We see that the norm of the solution increases exponentially, however *the dominant term is given by the projection onto the lowest frequency signal* and in fact

$$\frac{\mathrm{vec}(\mathbf{F}(t))}{\|\mathrm{vec}(\mathbf{F}(t))\|} = \frac{P^\perp_{\ker(\mathbf{\Delta})} \mathrm{vec}(\mathbf{F}(0)) + \mathcal{O}(e^{-\mathrm{gap}(\mathbf{\Delta})t})(\mathbf{I} - P^\perp_{\ker(\mathbf{\Delta})}) \mathrm{vec}(\mathbf{F}(0))}{\left( \|P^\perp_{\ker(\mathbf{\Delta})} \mathrm{vec}(\mathbf{F}(0))\|^2 + \mathcal{O}(e^{-2\mathrm{gap}(\mathbf{\Delta})t}) \right)^{\frac{1}{2}}} \to \mathrm{vec}(\mathbf{F}_\infty),$$

such that $(\mathbf{I}_d \otimes \mathbf{\Delta})\mathrm{vec}(\mathbf{F}_\infty) = \mathbf{0}$ which means $\mathbf{\Delta}\mathbf{f}^r_\infty = \mathbf{0}$, for each column $1 \le r \le d$. Equivalently, one can compute $\mathcal{E}^{\mathrm{Dir}}(\mathbf{F}(t)/\|\mathbf{F}(t)\|)$ and conclude that the latter quantity converges to zero as $t \to \infty$ by the very same argument.

In fact, this motivates further the nomenclature LFD and HFD. Without loss of generality we focus now on the high-frequency case. Assume that we have a HFD dynamics $t \mapsto \mathbf{F}(t)$, i.e. $\mathcal{E}^{\mathrm{Dir}}(\mathbf{F}(t)/\|\mathbf{F}(t)\|) \to \lambda_{n-1}$, then we can vectorize the solution and write $\mathrm{vec}(\mathbf{F}(t)) = \|\mathbf{F}(t)\|\mathrm{vec}(\mathbf{Q}(t))$, for some time-dependent unit vector $\mathrm{vec}(\mathbf{Q}(t)) \in \mathbb{R}^{nd}$:

$$\mathrm{vec}(\mathbf{Q}(t)) = \sum_{r,\ell} q_{r,\ell}(t) \mathbf{e}_r \otimes \boldsymbol{\phi}_\ell, \quad \sum_{r,\ell} q_{r,\ell}^2(t) = 1.$$

By Lemma B.1 and more explicitly (25), we derive that the coefficients $\{q_{r,\lambda_{n-1}}\}$ associated with the eigenvenctors $\mathbf{e}_r \otimes \boldsymbol{\phi}_{\lambda_{n-1}}$ are dominant in the evolution hence justifying the name *high-frequency dominated dynamics*.

The next result provides a theoretical justification for the characterization of low (high) frequency dominated dynamics in Definition 4.2.

**Lemma B.2.** *Consider a dynamical system $\dot{\mathbf{F}}(t) = \mathrm{GNN}_\theta(\mathbf{F}(t), t)$, with initial condition $\mathbf{F}(0)$.*

    *(i) $\mathrm{GNN}_\theta$ is LFD if and only if $(\mathbf{I}_d \otimes \mathbf{\Delta})\frac{\mathrm{vec}(\mathbf{F}(t))}{\|\mathbf{F}(t)\|} \to \mathbf{0}$ if and only if for each sequence $t_j \to \infty$ there exist a subsequence $t_{j_k} \to \infty$ and $\mathbf{F}_\infty$ (depending on the subsequence) s.t. $\frac{\mathbf{F}(t_{j_k})}{\|\mathbf{F}(t_{j_k})\|} \to \mathbf{F}_\infty$ satisfying $\mathbf{\Delta}\mathbf{f}^r_\infty = \mathbf{0}$, for each $1 \le r \le d$.*

*(ii)* GNN$_\theta$ *is* HFD *if and only if for each sequence* $t_j \to \infty$ *there exist a subsequence* $t_{j_k} \to \infty$ *and* $\mathbf{F}_\infty$ *(depending on the subsequence) s.t.* $\frac{\mathbf{F}(t_{j_k})}{\|\mathbf{F}(t_{j_k})\|} \to \mathbf{F}_\infty$ *satisfying* $\mathbf{\Delta f}^r_\infty = \lambda_{n-1}\mathbf{f}^r_\infty$, *for each* $1 \le r \le d$.

*Proof.* (i) Since $\mathbf{\Delta f}^r(t) \to \mathbf{0}$ for each $1 \le r \le d$ if and only if $(\mathbf{I}_d \otimes \mathbf{\Delta})\text{vec}(\mathbf{F}(t)) \to \mathbf{0}$, we conclude that the dynamics is LFD if and only if $(\mathbf{I}_d \otimes \mathbf{\Delta})\frac{\text{vec}(\mathbf{F}(t))}{\|\mathbf{F}(t)\|} \to \mathbf{0}$ due to (i) in Lemma B.1. Consider a sequence $t_j \to \infty$. Since $\text{vec}(\mathbf{F}(t_j))/\|\mathbf{F}(t_j)\|$ is a bounded sequence we can extract a converging subsequence $t_{j_k}$: $\text{vec}(\mathbf{F}(t_{j_k}))/\|\mathbf{F}(t_{j_k})\| \to \text{vec}(\mathbf{F}_\infty)$. If the dynamics is LFD, then $(\mathbf{I}_d \otimes \mathbf{\Delta})\frac{\text{vec}(\mathbf{F}(t_{j_k}))}{\|\mathbf{F}(t_{j_k})\|} \to \mathbf{0}$ and hence we conclude that $\text{vec}(\mathbf{F}_\infty) \in \ker(\mathbf{I}_d \otimes \mathbf{\Delta})$. Conversely, assume that for any sequence $t_j \to \infty$ there exists a subsequence $t_{j_k}$ and $\mathbf{F}_\infty$ such that $\frac{\mathbf{F}(t_{j_k})}{\|\mathbf{F}(t_{j_k})\|} \to \mathbf{F}_\infty$ satisfying $\mathbf{\Delta f}^r_\infty = \mathbf{0}$, for each $1 \le r \le d$. If for a contradiction we had $\varepsilon > 0$ and $t_j \to \infty$ such that $\mathcal{E}^{\text{Dir}}(\mathbf{F}(t_j)/\|\mathbf{F}(t_j)\|) \ge \varepsilon$ – for $j$ large enough – then by (i) in Lemma B.1 there exist $1 \le r \le d$, $\ell > 0$ and a subsequence $t_{j_k}$ satisfying

$$\left|\left\langle\left(\frac{\text{vec}(\mathbf{F}(t_{j_k}))}{\|\mathbf{F}(t_{j_k})\|}\right), \mathbf{e}_r \otimes \boldsymbol{\phi}_\ell\right\rangle\right| > \delta(\varepsilon) > 0,$$

meaning that there is no subsequence of $\{t_{j_k}\}$ s.t. $(\mathbf{I}_d \otimes \mathbf{\Delta})\text{vec}(\mathbf{F}(t_{j_k}))/\|\mathbf{F}(t_{j_k})\| \to \mathbf{0}$, providing a contradiction.

(ii) This is equivalent to (ii) in Lemma B.1.

$\square$

**Remark.** *We note that in Lemma B.2 an* LFD *dynamics does not necessarily mean that the normalized solution converges to the kernel of* $\mathbf{I}_d \otimes \mathbf{\Delta}$ *– i.e. one in general has always to pass to subsequences. Indeed, we can consider the simple example* $t \mapsto \text{vec}(\mathbf{F}(t)) := \cos(t)\mathbf{e}_{\bar{r}} \otimes \boldsymbol{\phi}_0$, *for some* $1 \le \bar{r} \le d$, *which satisfies* $\mathbf{\Delta f}^r(t) = \mathbf{0}$ *for each* $r$, *but it is not a convergent function due to its oscillatory nature. Same argument applies to* HFD.

We will now show that (8) can lead to a HFD dynamics. To this end, we assume that $\mathbf{\Omega} = \tilde{\mathbf{W}} = \mathbf{0}$ so that (8) becomes $\dot{\mathbf{F}}(t) = \mathbf{A}\mathbf{F}(t)\mathbf{W}$. According to (9) the negative eigenvalues of $\mathbf{W}$ lead to repulsion. We show that the latter can induce HFD dynamics as per Definition 4.2. We let $P_{\mathbf{W}}^{\rho_-}$ be the orthogonal projection into the eigenspace of $\mathbf{W} \otimes \mathbf{A}$ associated with the eigenvalue $\rho_- := |\mu_0|(\lambda_{n-1} - 1)$. We recall that $\mu_0$ is the most negative eigenvalue of $\mathbf{W}$, while $\mu_{d-1}$ is the most positive eigenvalue of $\mathbf{W}$. We define $\epsilon_{\text{HFD}}$ by:

$$\epsilon_{\text{HFD}} := \min\{\rho_- - \mu_{d-1}, |\mu_0|\text{gap}(\lambda_{n-1}\mathbf{I} - \mathbf{\Delta}), \text{gap}(|\mu_0|\mathbf{I} + \mathbf{W})(\lambda_{n-1} - 1)\}.$$

**Theorem B.3.** *If* $\rho_- > \mu_{d-1}$, *then* $\dot{\mathbf{F}}(t) = \mathbf{A}\mathbf{F}(t)\mathbf{W}$ *is* HFD *for a.e.* $\mathbf{F}(0)$ *and indeed we have*

$$\mathcal{E}^{\text{Dir}}(\mathbf{F}(t)) = e^{2t\rho_-}\left(\lambda_{n-1}\|P_{\mathbf{W}}^{\rho_-}\mathbf{F}(0)\|^2 + \mathcal{O}(e^{-2t\epsilon_{\text{HFD}}})\right), \quad t \ge 0,$$

*and* $\mathbf{F}(t)/\|\mathbf{F}(t)\|$ *converges to* $\mathbf{F}_\infty \in \mathbb{R}^{n \times d}$ *such that* $\mathbf{\Delta f}^r_\infty = \lambda_{n-1}\mathbf{f}^r_\infty$, *for* $1 \le r \le d$. *If instead* $\rho_- < \mu_{d-1}$ *then the dynamics is* LFD *and* $\mathbf{F}(t)/\|\mathbf{F}(t)\|$ *converges to* $\mathbf{F}_\infty \in \mathbb{R}^{n \times d}$ *such that* $\mathbf{\Delta f}^r_\infty = \mathbf{0}$, *for* $1 \le r \le d$, *exponentially fast.*

*Proof of Theorem B.3.* Once we compute the spectrum of $\mathbf{W} \otimes \mathbf{A}$ via (20), we can write the solution as – recall that $\mathbf{A} = \mathbf{I}_n - \mathbf{\Delta}$ so we can express the eigenvalues of $\mathbf{A}$ in terms of the eigenvalues of $\mathbf{\Delta}$:

$$\text{vec}(\mathbf{F}(t)) = \sum_{r,\ell} e^{\mu_r(1-\lambda_\ell)t}c_{r,\ell}(0)\boldsymbol{\psi}_r \otimes \boldsymbol{\phi}_\ell,$$

with $\mathbf{W}\boldsymbol{\psi}_r = \mu_r\boldsymbol{\psi}_r$, for $0 \le r \le d-1$, where $\{\boldsymbol{\psi}_r\}_r$ is an orthonormal basis of eigenvectors in $\mathbb{R}^d$. We can then calculate the Dirichlet energy along the solution as

$$\mathcal{E}^{\text{Dir}}(\mathbf{F}(t)) = \langle\text{vec}(\mathbf{F}(t)), (\mathbf{I}_d \otimes \mathbf{\Delta})\text{vec}(\mathbf{F}(t))\rangle = \sum_{r,\ell} e^{2\mu_r(1-\lambda_\ell)t}c_{r,\ell}^2(0)\lambda_\ell.$$

We now consider two cases:

- If $\mu_r > 0$, then $\mu_r(1 - \lambda_\ell) \leq \mu_{d-1}$.

- If $\mu_r < 0$, then $\mu_r(1 - \lambda_\ell) \leq |\mu_0|(\lambda_{n-1} - 1) := \rho_-$, with eigenvectors $\boldsymbol{\psi}_r \otimes \boldsymbol{\phi}_{n-1}$ for each $r$ s.t. $\mathbf{W}\boldsymbol{\psi}_r = \mu_0\boldsymbol{\psi}_r$ – without loss of generality we can assume that $\lambda_{n-1}$ is a simple eigenvalue for $\boldsymbol{\Delta}$. In particular, if $\mu_r < 0$ and $\mu_r(1 - \lambda_\ell) < \rho_-$, then

$$\mu_r(1 - \lambda_\ell) < \max\{|\mu_0|(\lambda_{n-2} - 1), |\mu_1|(\lambda_{n-1} - 1)\},$$

where $\mu_1$ is the second most negative eigenvalue of $\mathbf{W}$ and $\lambda_{n-2}$ is the second largest eigenvalue of $\boldsymbol{\Delta}$. In particular, we can write

$$\lambda_{n-2} = \lambda_{n-1} - \text{gap}(\lambda_{n-1}\mathbf{I}_n - \boldsymbol{\Delta}), \quad |\mu_1| = |\mu_0| - \text{gap}(|\mu_0|\mathbf{I}_d + \mathbf{W}). \tag{26}$$

From (i) and (ii) we derive that if $\mu_r(1 - \lambda_\ell) \neq \rho_-$, then

$$\begin{aligned}
\mu_r(1 - \lambda_\ell) - \rho_- &< -\min\{\rho_- - \mu_{d-1}, \rho_- - |\mu_0|(\lambda_{n-2} - 1), \rho_- - |\mu_1|(\lambda_{n-1} - 1)\} \\
&= -\min\{\rho_- - \mu_{d-1}, |\mu_0|\text{gap}(\lambda_{n-1}\mathbf{I} - \boldsymbol{\Delta}), \text{gap}(|\mu_0|\mathbf{I} + \mathbf{W})(\lambda_{n-1} - 1)\} = -\epsilon_{\text{HFD}},
\end{aligned} \tag{27}$$

where we have used (26). Accordingly, if $\rho_- > \mu_{d-1}$, then

$$\begin{aligned}
\mathcal{E}^{\text{Dir}}(\mathbf{F}(t)) &= e^{2t\rho_-}\left(\lambda_{n-1}\sum_{r:\mu_r=\mu_0} c^2_{r,\lambda_{n-1}}(0) + \sum_{r,\ell:\mu_r(1-\lambda_\ell)\neq\rho_-} e^{2(\mu_r(1-\lambda_\ell)-\rho_-)t}c^2_{r,\ell}(0)\right) \\
&= e^{2t\rho_-}\left(\lambda_{n-1}\|P^{\rho_-}_{\mathbf{W}}\mathbf{F}(0)\|^2 + \mathcal{O}(e^{-2t\epsilon_{\text{HFD}}})\right).
\end{aligned}$$

By the same argument we can factor out the dominant term and derive the following limit for $t \to \infty$ and for a.e. $\mathbf{F}(0)$ since $P^{\rho_-}_{\mathbf{W}}\text{vec}(\mathbf{F}(0)) = \mathbf{0}$ only if $\text{vec}(\mathbf{F}(0))$ belongs to a lower dimensional subspace of $\mathbb{R}^{nd}$:

$$\frac{\text{vec}(\mathbf{F}(t))}{\text{vec}(\mathbf{F}(t))} = \frac{P^{\rho_-}_{\mathbf{W}}\text{vec}(\mathbf{F}(0)) + \mathcal{O}(e^{-\epsilon_{\text{HFD}}t})((\mathbf{I} - P^{\rho_-}_{\mathbf{W}})\text{vec}(\mathbf{F}(0)))}{\left(\|P^{\rho_-}_{\mathbf{W}}\text{vec}(\mathbf{F}(0))\|^2 + \mathcal{O}(e^{-2\epsilon_{\text{HFD}}t})\right)^{\frac{1}{2}}} \to \frac{P^{\rho_-}_{\mathbf{W}}\text{vec}(\mathbf{F}(0))}{\|P^{\rho_-}_{\mathbf{W}}\text{vec}(\mathbf{F}(0))\|},$$

where the latter is a unit vector $\text{vec}(\mathbf{F}_\infty)$ satisfying $(\mathbf{I}_d \otimes \boldsymbol{\Delta})\text{vec}(\mathbf{F}_\infty) = \lambda_{n-1}\text{vec}(\mathbf{F}_\infty)$, which completes the proof for the HFD case.

For the opposite case the proof can be adapted without efforts as explicitly derived in the proof of Theorem 5.1. $\square$

We emphasize that the previous result includes Theorem 4.3 as a special case.

### B.3 Comparison with continuous GNNs: details and proofs

**Comparison with some continuous GNN models.** In contrast with Theorem 4.3, we show that three main *linearized* continuous GNN models are either *smoothing* or more generally LFD. The linearized PDE-GCN$_D$ model Eliasof et al. (2021) corresponds to choosing $\tilde{\mathbf{W}} = \mathbf{0}$ and $\boldsymbol{\Omega} = \mathbf{W} = \mathbf{K}(t)^\top\mathbf{K}(t)$ in (8), for some time-dependent family $t \mapsto \mathbf{K}(t) \in \mathbb{R}^{d\times d}$:

$$\dot{\mathbf{F}}_{\text{PDE-GCN}_{\text{D}}}(t) = -\boldsymbol{\Delta}\mathbf{F}(t)\mathbf{K}(t)^\top\mathbf{K}(t).$$

The CGNN model Xhonneux et al. (2020) can be derived from (8) by setting $\boldsymbol{\Omega} = \mathbf{I} - \tilde{\boldsymbol{\Omega}}, \mathbf{W} = -\tilde{\mathbf{W}} = \mathbf{I}$:

$$\dot{\mathbf{F}}_{\text{CGNN}}(t) = -\boldsymbol{\Delta}\mathbf{F}(t) + \mathbf{F}(t)\tilde{\boldsymbol{\Omega}} + \mathbf{F}(0).$$

Finally, in linearized GRAND Chamberlain et al. (2021a) a row-stochastic matrix $\boldsymbol{\mathcal{A}}(\mathbf{F}(0))$ is *learned* from the encoding via an attention mechanism and we have

$$\dot{\mathbf{F}}_{\text{GRAND}}(t) = -\boldsymbol{\Delta}_{\text{RW}}\mathbf{F}(t) = -(\mathbf{I} - \boldsymbol{\mathcal{A}}(\mathbf{F}(0)))\mathbf{F}(t).$$

We note that if $\boldsymbol{\mathcal{A}}$ is not symmetric, then GRAND is *not* a gradient flow.

**Theorem B.4.** $\mathrm{PDE}-\mathrm{GCN}_D$, CGNN *and* GRAND *satisfy the following:*

(i) $\mathrm{PDE}-\mathrm{GCN}_D$ *is a smoothing model:* $\dot{\mathcal{E}}^{\mathrm{Dir}}(\mathbf{F}_{\mathrm{PDE-GCN}_D}(t)) \leq 0$.

(ii) *For a.e.* $\mathbf{F}(0)$ *it holds:* CGNN *is never* HFD *and if we remove the source term, then* $\mathcal{E}^{\mathrm{Dir}}(\mathbf{F}_{\mathrm{CGNN}}(t)/\|\mathbf{F}_{\mathrm{CGNN}}(t)\|) \leq e^{-\mathrm{gap}(\boldsymbol{\Delta})t}$.

(iii) *If* $\mathsf{G}$ *is connected with self-loops,* $\mathbf{F}_{\mathrm{GRAND}}(t) \to \boldsymbol{\mu}$ *as* $t \to \infty$, *with* $\boldsymbol{\mu}^r = \mathrm{mean}(\mathbf{f}^r(0))$, $1 \leq r \leq d$.

By (ii) the source-free CGNN-evolution is LFD *independent of* $\tilde{\boldsymbol{\Omega}}$. Moreover, by (iii), over-smoothing occurs for GRAND. On the other hand, Theorem 4.3 shows that the negative eigenvalues of $\mathbf{W}$ can make the source-free gradient flow in (8) HFD. Experiments in Appendix E confirm that the gradient flow model outperforms CGNN and GRAND on heterophilic graphs.

We prove the following result which covers Theorem 5.4.

*Proof of Theorem B.4.* We structure the proof by following the numeration in the statement.

(i) From direct computation we find

$$\frac{d\mathcal{E}^{\mathrm{Dir}}(\mathbf{F}(t))}{dt} = \frac{d}{dt}\left(\langle\mathrm{vec}(\mathbf{F}(t)), (\mathbf{I}_d \otimes \boldsymbol{\Delta})\mathrm{vec}(\mathbf{F}(t))\rangle\right)$$
$$= -\langle\mathrm{vec}(\mathbf{F}(t)), (\mathbf{K}^\top(t)\mathbf{K}(t) \otimes \boldsymbol{\Delta}^2)\mathrm{vec}(\mathbf{F}(t))\rangle \leq 0,$$

since $\mathbf{K}^\top(t)\mathbf{K}(t) \otimes \boldsymbol{\Delta}^2 \succeq 0$. Note that we have used that $(\mathbf{A} \otimes \mathbf{B})(\mathbf{C} \otimes \mathbf{D}) = \mathbf{AC} \otimes \mathbf{BD}$.

(ii) We consider the dynamical system

$$\dot{\mathbf{F}}_{\mathrm{CGNN}}(t) = -\boldsymbol{\Delta}\mathbf{F}(t) + \mathbf{F}(t)\tilde{\boldsymbol{\Omega}} + \mathbf{F}(0).$$

We can write $\mathrm{vec}(\mathbf{F}(t)) = \sum_{r,\ell} c_{r,\ell}(t)\boldsymbol{\phi}_r^{\tilde{\boldsymbol{\Omega}}} \otimes \boldsymbol{\phi}_\ell$, leading to the system

$$\dot{c}_{r,\ell}(t) = (\lambda_r^{\tilde{\boldsymbol{\Omega}}} - \lambda_\ell)c_{r,\ell}(t) + c_{r,\ell}(0), \quad 0 \leq \ell \leq n-1, \ 1 \leq r \leq d.$$

We can solve explicitly the system as

$$c_{r,\ell}(t) = c_{r,\ell}(0)\left(e^{(\lambda_r^{\tilde{\boldsymbol{\Omega}}} - \lambda_\ell)t}\left(1 + \frac{1}{\lambda_r^{\tilde{\boldsymbol{\Omega}}} - \lambda_\ell}\right) - \frac{1}{\lambda_r^{\tilde{\boldsymbol{\Omega}}} - \lambda_\ell}\right), \quad \text{if } \lambda_r^{\tilde{\boldsymbol{\Omega}}} \neq \lambda_\ell$$
$$c_{r,\ell}(t) = c_{r,\ell}(0)(1 + t), \quad \text{otherwise.}$$

We see now that for a.e. $\mathbf{F}(0)$ the projection $(\mathbf{I}_d \otimes \boldsymbol{\phi}_{\lambda_{n-1}}(\boldsymbol{\phi}_{\lambda_{n-1}})^\top)\mathrm{vec}(\mathbf{F}(t))$ is never the dominant term. In fact, if there exists $r$ s.t. $\lambda_r^{\tilde{\boldsymbol{\Omega}}} \geq \lambda_{n-1}$, then $\lambda_r^{\tilde{\boldsymbol{\Omega}}} - \lambda_\ell > \lambda_r^{\tilde{\boldsymbol{\Omega}}} - \lambda_{n-1}$, for any other non-maximal graph Laplacian eigenvalue. It follows that there is *no* $\tilde{\boldsymbol{\Omega}}$ s.t. the normalized solution maximizes the Rayleigh quotient of $\mathbf{I}_d \otimes \boldsymbol{\Delta}$, proving that CGNN is never HFD.

If we have no source, then the CGNN equation becomes

$$\dot{\mathbf{F}}(t) = -\boldsymbol{\Delta}\mathbf{F}(t) + \mathbf{F}(t)\tilde{\boldsymbol{\Omega}} \iff \mathrm{vec}(\dot{\mathbf{F}}(t)) = (\tilde{\boldsymbol{\Omega}} \oplus (-\boldsymbol{\Delta}))\mathrm{vec}(\mathbf{F}(t)),$$

using the Kronecker sum notation in (21). It follows that we can write the vectorized solution in the basis $\{\boldsymbol{\phi}_r^{\tilde{\boldsymbol{\Omega}}} \otimes \boldsymbol{\phi}_\ell\}_{r,\ell}$ as

$$\mathrm{vec}(\mathbf{F}(t)) = e^{\lambda_+^{\tilde{\boldsymbol{\Omega}}}t}\left(\sum_{r:\lambda_r^{\tilde{\boldsymbol{\Omega}}}=\lambda_+^{\tilde{\boldsymbol{\Omega}}}} c_{r,0}(0)\boldsymbol{\phi}_r^{\tilde{\boldsymbol{\Omega}}} \otimes \boldsymbol{\phi}_0 + \mathcal{O}(e^{-\mathrm{gap}(\lambda_+^{\tilde{\boldsymbol{\Omega}}}\mathbf{I}_d - \tilde{\boldsymbol{\Omega}})t}) \sum_{r:\lambda_r^{\tilde{\boldsymbol{\Omega}}}<\lambda_+^{\tilde{\boldsymbol{\Omega}}}} c_{r,0}(0)\boldsymbol{\phi}_r^{\tilde{\boldsymbol{\Omega}}} \otimes \boldsymbol{\phi}_0\right)$$
$$+ e^{\lambda_+^{\tilde{\boldsymbol{\Omega}}}t}\left(\mathcal{O}(e^{-\mathrm{gap}(\boldsymbol{\Delta})t})\left(\sum_{r,\ell>0} c_{r,\ell}(0)\boldsymbol{\phi}_r^{\tilde{\boldsymbol{\Omega}}} \otimes \boldsymbol{\phi}_\ell\right)\right),$$

meaning that the dominant term is given by the lowest frequency component and in fact, if we normalize we find $\mathcal{E}^{\mathrm{Dir}}(\mathbf{F}(t)/\|\mathbf{F}(t)\|) \leq e^{-\mathrm{gap}(\boldsymbol{\Delta})t}$.

(iii) Finally, we consider the dynamical system induced by linear GRAND

$$\dot{\mathbf{F}}_{\mathrm{GRAND}}(t) = -\boldsymbol{\Delta}_{\mathrm{RW}}\mathbf{F}(t) = -(\mathbf{I} - \boldsymbol{\mathcal{A}}(\mathbf{F}(0)))\mathbf{F}(t).$$

Since we have no channel-mixing, without loss of generality we can assume that $d = 1$ – one can then extend the argument to any entry. We can use the Jordan form of $\boldsymbol{\mathcal{A}}$ to write the solution of the GRAND dynamical system as

$$\mathbf{f}(t) = P\mathrm{diag}(e^{J_1 t}, \ldots, e^{J_n t})P^{-1}\mathbf{f}(0),$$

for some invertible matrix $P$ of eigenvectors, with

$$e^{J_k t} = e^{-(1-\lambda_k^{\boldsymbol{\mathcal{A}}})t}\begin{pmatrix} 1 & t & \cdots & \frac{t^{m_k-1}}{(m_k-1)!} \\ & & & \vdots \\ & & & 1 \end{pmatrix},$$

where $m_k$ are the eigenvalue multiplicities. Since by assumption $\mathsf{G}$ is connected and augmented with self-loops, the row-stochastic attention matrix $\boldsymbol{\mathcal{A}}$ computed in Chamberlain et al. (2021a) with softmax activation is *regular*, meaning that there exists $m \in \mathbb{N}$ such that $(\boldsymbol{\mathcal{A}}^m)_{ij} > 0$ for each entry $(i,j)$. Accordingly, we can apply Perron Theorem to derive that any eigenvalue of $\boldsymbol{\mathcal{A}}$ has real part smaller than one except the eigenvalue $\lambda_0^{\boldsymbol{\mathcal{A}}}$ with multiplicity one, associated with the Perron eigenvector $\mathbf{1}_n$. Accordingly, we find that each block $e^{J_k t}$ decays to zero as $t \to \infty$ with the exception of the one $e^{J_0 t}$ associated with the Perron eigenvector. In particular, the projection of $\mathbf{f}_0$ over the Perron eigenvector is just $\mu \mathbf{1}_n$, with $\mu$ the average of the feature initial condition. This completes the proof. $\qquad\square$

### B.4 Propagating with the Laplacian

In this subsection we briefly review the special case of (8) where $\boldsymbol{\Omega} = \mathbf{W}$, and comment on why we generally expect a framework where the propagation is governed by $\mathbf{A}$ to be more flexible than one with $-\boldsymbol{\Delta}$. If $\boldsymbol{\Omega} = \mathbf{W}$ and we suppress the source term i.e. $\tilde{\mathbf{W}} = \mathbf{0}$, the gradient flow in (8) becomes

$$\dot{\mathbf{F}}(t) = -\boldsymbol{\Delta}\mathbf{F}(t)\mathbf{W}. \tag{28}$$

We note that once vectorized, the solution to the dynamical system can be written as

$$\mathrm{vec}(\mathbf{F}(t)) = \sum_{r=0}^{d-1}\sum_{\ell=0}^{n-1} e^{-\mu_r \lambda_\ell t}c_{r,\ell}(0)\boldsymbol{\psi}_r \otimes \boldsymbol{\phi}_\ell.$$

In particular, we immediately deduce the following counterpart to Theorem 4.3:

**Corollary B.5.** *If* $\mathrm{spec}(\mathbf{W}) \cap \mathbb{R}_- \neq \emptyset$*, then (28) is* HFD *for a.e.* $\mathbf{F}(0)$.

Differently from (8) the lowest frequency component is *always preserved independent of the spectrum of* $\mathbf{W}$. This means that the system cannot learn eigenvalues of $\mathbf{W}$ to either magnify or suppress the low-frequency projection. In contrast, this can be done if $\boldsymbol{\Omega} = \mathbf{0}$, or equivalently one replaces $-\boldsymbol{\Delta}$ with $\mathbf{A}$ providing a further *justification in terms of the interaction between graph spectrum and channel-mixing spectrum for why graph-convolutional models use the normalized adjacency rather than the Laplacian for propagating messages* Kipf & Welling (2017).

### B.5 Revisiting the connection with the manifold case

In (18) a constant nontrivial metric $h$ in $\mathbb{R}^d$ leads to the mixing of the feature channels. We adapt this idea by considering a symmetric positive semi-definite $\mathbf{H} = \mathbf{W}^\top\mathbf{W}$ with $\mathbf{W} \in \mathbb{R}^{d \times d}$ and using it to generalize $\mathcal{E}^{\mathrm{Dir}}$ by suitably weighting the norm of the edge gradients as

$$\mathcal{E}_{\mathbf{W}}^{\mathrm{Dir}}(\mathbf{F}) := \frac{1}{2} \sum_{q,r=1}^{d} \sum_{i} \sum_{j:(i,j)\in\mathsf{E}} h_{qr}(\nabla \mathbf{f}^q)_{ij} (\nabla \mathbf{f}^r)_{ij} = \frac{1}{2} \sum_{(i,j)\in\mathsf{E}} \|\mathbf{W}(\nabla\mathbf{F})_{ij}\|^2. \tag{29}$$

We note the analogy with (18), where the sum over the nodes replaces the integration over the domain and the $j$-th derivative at some point $i$ is replaced by the gradient along the edge $(i,j) \in \mathsf{E}$. We generally treat $\mathbf{W}$ as *learnable weights* and study the gradient flow of $\mathcal{E}_{\mathbf{W}}^{\mathrm{Dir}}$:

$$\dot{\mathbf{F}}(t) = -\frac{1}{2} \nabla_{\mathbf{F}} \mathcal{E}_{\mathbf{W}}^{\mathrm{Dir}}(\mathbf{F}(t)) = -\mathbf{\Delta}\mathbf{F}(t)\mathbf{W}^{\top}\mathbf{W}. \tag{30}$$

We see that (30) generalizes (3).

**Proposition B.6.** *Let $P_{\mathbf{W}}^{\mathrm{ker}}$ be the projection onto $\ker(\mathbf{W}^{\top}\mathbf{W})$. Equation (30) is smoothing since*

$$\mathcal{E}^{\mathrm{Dir}}(\mathbf{F}(t)) \leq e^{-2t\mathrm{gap}(\mathbf{W}^{\top}\mathbf{W})\mathrm{gap}(\mathbf{\Delta})} \|\mathbf{F}(0)\|^2 + \mathcal{E}^{\mathrm{Dir}}((P_{\mathbf{W}}^{\mathrm{ker}} \otimes \mathbf{I}_n)\mathrm{vec}(\mathbf{F}(0))), \quad t \geq 0.$$

*In fact $\mathbf{F}(t) \to \mathbf{F}_{\infty}$ s.t. $\exists \, \phi_{\infty} \in \mathbb{R}^d$: for each $i \in \mathsf{V}$ we have $(\mathbf{f}_{\infty})_i = \sqrt{\frac{d_i}{2|\mathsf{E}|}} \phi_{\infty} + P_{\mathbf{W}}^{\mathrm{ker}} \mathbf{f}_i(0)$.*

*Proof of Proposition B.6.* We can vectorize the gradient flow system in (30) and use the spectral characterization of $\mathbf{W}^{\top}\mathbf{W} \otimes \mathbf{\Delta}$ in (20) to write the solution explicitly as

$$\mathrm{vec}(\mathbf{F}(t)) = \sum_{r,\ell} e^{-(\mu_r \lambda_\ell)t} c_{r,\ell}(0) \boldsymbol{\psi}_r \otimes \boldsymbol{\phi}_\ell,$$

where $\{\mu_r\}_r = \mathrm{spec}(\mathbf{W}^{\top}\mathbf{W}) \subset \mathbb{R}_{\geq 0}$ with associated basis of orthonormal eigenvectors given by $\{\boldsymbol{\psi}_r\}_r$. Then

$$\mathcal{E}^{\mathrm{Dir}}(\mathbf{F}(t)) = \langle \mathrm{vec}(\mathbf{F}(t)), (\mathbf{I}_d \otimes \mathbf{\Delta})\mathrm{vec}(\mathbf{F}(t)) \rangle = \sum_{r,\ell} e^{-2t(\mu_r \lambda_\ell)} c_{r,\ell}^2(0) \lambda_\ell$$

$$= \sum_{r:\mu_r=0,\ell} c_{r,\ell}^2(0)\lambda_\ell + \sum_{r:\mu_r>0,\ell>0} c_{r,\ell}^2(0) e^{-2t(\mu_r \lambda_\ell)} \lambda_\ell$$

$$= \mathcal{E}^{\mathrm{Dir}}((P_{\mathbf{W}}^{\mathrm{ker}} \otimes \mathbf{I}_n)\mathrm{vec}(\mathbf{F}(0))) + \sum_{r:\mu_r>0,\ell>0} c_{r,\ell}^2(0) e^{-2t(\mu_r \lambda_\ell)} \lambda_\ell$$

$$\leq \mathcal{E}^{\mathrm{Dir}}((P_{\mathbf{W}}^{\mathrm{ker}} \otimes \mathbf{I}_n)\mathrm{vec}(\mathbf{F}(0))) + \lambda_{n-1} e^{-2t\mathrm{gap}(\mathbf{W}^{\top}\mathbf{W})\mathrm{gap}(\mathbf{\Delta})} \|\mathbf{F}(0)\|^2,$$

where we recall that $P_{\mathbf{W}}^{\mathrm{ker}}$ is the projection onto $\ker(\mathbf{W}^{\top}\mathbf{W})$ and that by convention the index $\ell = 0$ is associated with the lowest graph frequency $\lambda_0 = 0$ – by assumption $\mathsf{G}$ is connected. This proves that the dynamics is in fact smoothing. By the very same argument we find that

$$\mathrm{vec}(\mathbf{F}(t)) \to (\mathbf{I}_d \otimes P^{\mathrm{ker}})\mathrm{vec}(\mathbf{F}(0)) + (P_{\mathbf{W}}^{\mathrm{ker}} \otimes \mathbf{I}_n)\mathrm{vec}(\mathbf{F}(0)), \quad t \to \infty,$$

with $P^{\mathrm{ker}}$ the orthogonal projection onto $\ker\mathbf{\Delta}$ – the other terms decay exponentially to zero. We first focus on the first quantity, which we can write as

$$(\mathbf{I}_d \otimes P^{\mathrm{ker}})\mathrm{vec}(\mathbf{F}(0)) = \sum_r c_{r,0}(0) \boldsymbol{\psi}_r \otimes \boldsymbol{\phi}_0,$$

which has matrix representation $\boldsymbol{\phi}_0 \boldsymbol{\phi}_{\infty}^{\top} \in \mathbb{R}^{n \times d}$ with

$$\boldsymbol{\phi}_{\infty} := \sum_r c_{r,0}(0) \boldsymbol{\psi}_r.$$

By (23) we deduce that the $i$-th row of $\boldsymbol{\phi}_0 \boldsymbol{\phi}_{\infty}^{\top} \in \mathbb{R}^{n \times d}$ is the $d$-dimensional vector $\sqrt{\frac{d_i}{2|\mathsf{E}|}} \boldsymbol{\phi}_{\infty}$. We now focus on the term

$$(P_{\mathbf{W}}^{\mathrm{ker}} \otimes \mathbf{I}_n)\mathrm{vec}(\mathbf{F}(0)) = \sum_{r:\mu_r=0,j} c_{r,j}(0) \boldsymbol{\psi}_r \otimes \boldsymbol{\phi}_j$$

which has matrix representation $\sum_{r:\mu_r=0,j} c_{r,j}(0)\phi_j(\psi_r)^\top$. In particular, the $i$-th row is given by

$$\sum_{r:\mu_r=0,j} c_{r,j}(0)(\phi_j)_i\psi_r = P_{\mathbf{W}}^{\mathrm{ker}}\mathbf{f}_i(0).$$

This completes the proof of Proposition B.6. $\qquad\square$

Proposition B.6 implies that no weight matrix $\mathbf{W}$ in (30) **can separate the limit embeddings $\mathbf{F}_\infty$ of nodes with same degree and same input features**. In particular, we have the following characterization:

- Projections of the edge gradients $(\nabla\mathbf{F})_{ij}(0) \in \mathbb{R}^d$ into the eigenvectors of $\mathbf{W}^\top\mathbf{W}$ with positive eigenvalues *shrink* along the GNN and converge to zero exponentially fast as integration time (depth) increases.

- Projections of the edge gradients $(\nabla\mathbf{F})_{ij}(0) \in \mathbb{R}^d$ into the kernel of $\mathbf{W}^\top\mathbf{W}$ stay *invariant*.

If $\mathbf{W}$ has a trivial kernel, then nodes with same degrees converge to the same representation hence incurring *over-smoothing*. Differently from Nt & Maehara (2019); Oono & Suzuki (2020); Cai & Wang (2020), over-smoothing occurs independently of the spectral radius of the 'channel-mixing' if its eigenvalues are *positive* – **even for equations which lead to residual** GNNs when discretized (Chen et al., 2018). According to Proposition B.6, we do not expect (30) to succeed on heterophilic graphs where *smoothing* processes are generally harmful. To deal with heterophily, one needs negative eigenvalues to generate repulsive forces among adjacent features.

## C  Proofs and additional details of Section 5

We first explicitly report here the expansion of the discrete gradient flow in (10) after $m$ layers to further highlight how this is not equivalent to a single linear layer with a message passing matrix $\mathbf{A}^m$ as for SGCN Wu et al. (2019). For simplicity we suppress the source term.

$$\mathbf{F}(t+\tau) = \mathbf{F}(t) + \tau\left(-\mathbf{F}(t)\mathbf{\Omega} + \mathbf{A}\mathbf{F}(t)\mathbf{W}\right)$$
$$\mathrm{vec}(\mathbf{F}(t+\tau)) = \left(\mathbf{I}_{nd} + \tau\left(-\mathbf{\Omega}\otimes\mathbf{I}_n + \mathbf{W}\otimes\mathbf{A}\right)\right)\mathrm{vec}(\mathbf{F}(t))$$
$$\mathrm{vec}(\mathbf{F}(m\tau)) = \sum_{k=0}^m \binom{m}{k}\tau^k\left(-\mathbf{\Omega}\otimes\mathbf{I}_n + \mathbf{W}\otimes\mathbf{A}\right)^k\mathrm{vec}(\mathbf{F}(0)) \tag{31}$$

and we see how the message passing matrix $\mathbf{A}$ actually enters the expansion after $m$ layers with each power $0 \le k \le m$. This is not surprising, given that *we are discretizing a linear dynamical system, meaning that we are approximating an exponential matrix*.

### C.1  From energy to evolution equations: exact expansion of the GNN solutions

We first address the proof of the main result.

*Proof of Theorem 5.1.* We consider a linear dynamical system

$$\mathbf{F}(t+\tau) = \mathbf{F}(t) + \tau\mathbf{A}\mathbf{F}(t)\mathbf{W},$$

with $\mathbf{W}$ symmetric. We vectorize the system and rewrite it as

$$\mathrm{vec}(\mathbf{F}(t+\tau)) = (\mathbf{I}_{nd} + \tau\mathbf{W}\otimes\mathbf{A})\mathrm{vec}(\mathbf{F}(t))$$

which in particular leads to

$$\mathrm{vec}(\mathbf{F}(m\tau)) = (\mathbf{I}_{nd} + \tau\mathbf{W}\otimes\mathbf{A})^m\mathrm{vec}(\mathbf{F}(0)).$$

We can then write explicitly the solution as

$$\text{vec}(\mathbf{F}(m\tau)) = \sum_{r,\ell} (1 + \tau\mu_r(1-\lambda_\ell))^m \, c_{r,\ell}(0)\boldsymbol{\psi}_r \otimes \boldsymbol{\phi}_\ell.$$

We now verify that by assumption in (12) the dominant term of the solution is the projection into the eigenspace associated with the eigenvalue $\rho_- = |\mu_0|(\lambda_{n-1}-1)$. The following argument follows the same structure in the proof of Theorem B.3 with the extra condition given by the step-size. First, we note that for any $r$ such that $\mu_r > 0$, we have

$$|1 + \tau\rho_-| > |1 + \tau\mu_{d-1}| \ge |1 + \tau\mu_r(1-\lambda_\ell)|$$

since we required $\rho_- > \mu_{d-1}$ in (12). Conversely, if $\mu_r < 0$, then

$$|1 + \tau\mu_r(1-\lambda_\ell)| \le \max\{|1+\tau\rho_-|, |1+\tau\mu_0|\}$$

Assume that $\tau|\mu_0| > 1$, otherwise there is nothing to prove. Then $|1+\tau\rho_-| > \tau|\mu_0| - 1$ if and only if

$$\tau|\mu_0|(2-\lambda_{n-1}) < 2,$$

which is precisely the right inequality in (12). We can then argue exactly as in the proof of Theorem B.3 to derive that for each index $r$ such that $\mu_r < 0$ and $\mu_r \ne \mu_0$, then

$$|1 + \tau\mu_r(1-\lambda_\ell)| \le \max\{|1+\tau|\mu_1|(\lambda_{n-1}-1)|, |1+\tau|\mu_0|(\lambda_{n-2}-1)|\}$$

with $\mu_1$ and $\lambda_{n-2}$ defined in (26). We can then introduce

$$\delta_{\text{HFD}} := \max\{\mu_{d-1}, \rho_- - |\mu_0|\text{gap}(\lambda_{n-1}\mathbf{I} - \boldsymbol{\Delta}), \rho_- - (\lambda_{n-1}-1)\text{gap}(|\mu_0|\mathbf{I} + \mathbf{W}), |\mu_0| - \frac{2}{\tau}\} \tag{32}$$

and conclude that

$$\mathbf{f}_i(m\tau) = \sum_{r,\ell} (1 + \tau\mu_r(1-\lambda_\ell))^m \, c_{r,\ell}(0)\boldsymbol{\phi}_\ell(i)\boldsymbol{\psi}_r$$

$$= (1+\tau\rho_-)^m \Big(c_{0,n-1}(0)\boldsymbol{\phi}_{n-1}(i)\cdot\boldsymbol{\psi}_0 + \mathcal{O}\Big(\Big(\frac{1+\tau\delta_{\text{HFD}}}{1+\tau\rho_-}\Big)^m\Big) \sum_{\ell,r:\mu_r(1-\lambda_\ell)\ne\rho_-} c_{r,\ell}(0)\boldsymbol{\phi}_\ell(i)\boldsymbol{\psi}_r\Big)$$

$$= (1+\tau\rho_-)^m \left(c_{0,n-1}(0)\boldsymbol{\phi}_{n-1}(i)\cdot\boldsymbol{\psi}_0 + \mathcal{O}(\delta^m)\right),$$

which completes the proof of (13).

Conversely, if $\rho_- < \mu_{d-1}$, then the projection onto the eigenspace spanned by $\boldsymbol{\psi}_{d-1}\otimes\boldsymbol{\phi}_0$ is dominating the dynamics with exponential growth $(1+\tau\mu_{d-1}(1+0))^m$. We can then adapt the very same argument above by factoring out the dominating term once we note that due to the choice of symmetric normalized Laplacian $\boldsymbol{\Delta}$, we have $\boldsymbol{\phi}_0(i) = \sqrt{d_i/2|\mathsf{E}|}$, which then yields (14).

$\square$

We can now also address the proof of Corollary 5.2.

*Proof of Corollary 5.2.* Once we have the node-wise expansion we can simply compute the Rayleigh quotient of $\mathbf{I}_d\otimes\boldsymbol{\Delta}$. We report the explicit details for the HFD case since the argument for LFD extends without relevant modifications. Using (11), we can compute the Dirichlet energy along a solution of $\mathbf{F}(t+\tau) = \mathbf{F}(t)+\tau\mathbf{A}\mathbf{F}(t)\mathbf{W}$ satisfying (12) by

$$\mathcal{E}^{\text{Dir}}(\mathbf{F}(m\tau)) = \sum_{r,\ell} (1+\tau\mu_r(1-\lambda_\ell))^{2m} c_{r,\ell}^2(0)\lambda_\ell$$

$$= (1+\tau\rho_-)^{2m}\Big(\lambda_{n-1}\sum_{r:\mu_r=\mu_0} c_{r,\lambda_{n-1}}^2(0) + \mathcal{O}\Big(\Big(\frac{1+\tau\delta_{\text{HFD}}}{1+\tau\rho_-}\Big)\Big)^{2m} \sum_{\ell,r:\mu_r(1-\lambda_\ell)\ne\rho_-} c_{r,\ell}^2(0)\lambda_\ell\Big)$$

$$= (1+\tau\rho_-)^{2m}\Big(\lambda_{n-1}\|P_{\mathbf{W}}^{\rho_-}\mathbf{F}(0)\|^2 + \mathcal{O}\Big(\Big(\frac{1+\tau\delta_{\text{HFD}}}{1+\tau\rho_-}\Big)^{2m}\Big)\Big),$$

where $P_{\mathbf{W}}^{\rho_-}$ is the orthogonal projector onto the eigenspace associated with the eigenvalue $\rho_- = |\mu_0|(\lambda_{n-1} - 1)$. In particular, since

$$\mathrm{vec}(\mathbf{F}(m\tau)) = (1 + \tau\rho_-)^m \left( P_{\mathbf{W}}^{\rho_-}\mathrm{vec}(\mathbf{F}(0)) + \mathcal{O}(\delta^m) \right),$$

we find that the dynamics is HFD with $\mathrm{vec}(\mathbf{F}(t))/\|\mathrm{vec}(\mathbf{F}(t))\|$ converging to the unit projection of the initial projection by $P_{\mathbf{W}}^{\rho_-}$ provided that such projection is not zero, which is satisfied for a.e. initial condition $\mathbf{F}(0)$.

$\square$

*Proof of Theorem 5.3.* If we drop the residual connection and simply consider $\mathbf{F}(t+\tau) = \tau\mathbf{A}\mathbf{F}(t)\mathbf{W}$, then

$$\mathrm{vec}(\mathbf{F}(m\tau)) = (\tau\mathbf{W} \otimes \mathbf{A})^m\mathrm{vec}(\mathbf{F}(0)).$$

Since $\mathsf{G}$ is not bipartite, the Laplacian spectral radius satisfies $\lambda_{n-1} < 2$. Therefore, for each pair of indices $(r, \ell)$ we have the following bound:

$$|\mu_r(1 - \lambda_\ell)| \leq \max\{\mu_{d-1}, |\mu_0|\},$$

and the inequality becomes strict if $\ell > 0$, i.e. $\lambda_\ell > 0$. The eigenvalues $\mu_{d-1}$ and $\mu_0$ are attained along the eigenvectors $\boldsymbol{\psi}_{d-1} \otimes \boldsymbol{\phi}_0$ and $\boldsymbol{\psi}_0 \otimes \boldsymbol{\phi}_0$ respectively. Accordingly, the dominant terms of the evolution lie in the kernel of $\mathbf{I}_d \otimes \boldsymbol{\Delta}$, meaning that for any $\mathbf{F}_0$ with non-zero projection in $\ker(\mathbf{I}_d \otimes \boldsymbol{\Delta})$ – which is satisfied by all initial conditions except those belonging to a lower dimensional subspace – the dynamics is LFD. In fact, without loss of generality assume that $|\mu_0| > \mu_{d-1}$, then

$$\begin{aligned}
\mathrm{vec}(\mathbf{F}(m\tau)) = {} & |\mu_0|^m \sum_{r:\mu_r=\mu_0} (-1)^m c_{r,0}(0)\boldsymbol{\psi}_0 \otimes \boldsymbol{\phi}_0 \\
& + |\mu_0|^m \Big( \mathcal{O}(\varphi(m)) \Big( \mathbf{I}_{nd} - \sum_{r:\mu_r=\mu_0} (\boldsymbol{\psi}_0 \otimes \boldsymbol{\phi}_0)(\boldsymbol{\psi}_0 \otimes \boldsymbol{\phi}_0)^\top \Big)\mathrm{vec}(\mathbf{F}(0)) \Big),
\end{aligned}$$

with $\varphi(m) \to 0$ as $m \to \infty$, which completes the proof. $\square$

**Gradient flow as spectral GNNs.** We finally discuss (10) from the perspective of spectral GNNs as in Balcilar et al. (2020). Let us assume that $\tilde{\mathbf{W}} = \mathbf{0}$, $\boldsymbol{\Omega} = \mathbf{0}$. If we let $\boldsymbol{\Delta} = \boldsymbol{\Psi}\mathrm{diag}(\boldsymbol{\mu})\boldsymbol{\Psi}^\top$ be the eigendecomposition of the graph Laplacian and $\{\mu_r\}$ be the spectrum of $\mathbf{W}$ with associated orthonormal basis of eigenvectors given by $\{\boldsymbol{\psi}_r\}$, and we introduce $\mathbf{z}^r(t) : \mathsf{V} \to \mathbb{R}$ defined by $z_i^r(t) = \langle \mathbf{f}_i(t), \boldsymbol{\psi}_r \rangle$, then we can rewrite the discretized gradient flow as

$$\mathbf{z}^r(t+\tau) = \boldsymbol{\Psi}(\mathbf{I} + \tau\mu_r(\mathbf{I} - \boldsymbol{\Lambda}^{\boldsymbol{\Delta}}))\boldsymbol{\Psi}^\top \mathbf{z}^r(t) = \mathbf{z}^r(t) + \tau\mu_r\mathbf{A}\mathbf{z}^r(t), \quad 0 \leq r \leq d-1. \tag{33}$$

Accordingly, for each projection into the $r$-th eigenvector of $\mathbf{W}$, we have a spectral function in the graph frequency domain given by $\lambda \mapsto 1 + \tau\mu_r(1 - \lambda)$. If $\mu_r > 0$ we have a *low-pass* filter while if $\mu_r < 0$ we have a *high-pass* filter. Moreover, we see that along the eigenvectors of $\mathbf{W}$, if $\mu_r < 0$ then the dynamics is equivalent to flipping the sign of the edge weights, which offers a direct comparison with methods proposed in Bo et al. (2021); Yan et al. (2021) where some 'attentive' mechanism is proposed to learn negative edge weights based on feature information.

The previous equation simply follows from

$$\begin{aligned}
z_i^r(t+\tau) = \langle \mathbf{f}_i(t+\tau), \boldsymbol{\psi}_r \rangle &= \langle \mathbf{f}_i(t) + \mathbf{W}(\mathbf{A}\mathbf{f}(t))_i, \boldsymbol{\psi}_r \rangle \\
&= z_i^r(t) + \mu_r \sum_j \bar{a}_{ij} z_j^r(t),
\end{aligned}$$

which concludes the derivation of (33).

# D  Proofs and additional details of Section 6

*Proof of Theorem 6.1.* First we check that if time is continuous, then $\mathcal{E}_\theta$ in (6) is decreasing. We use the Kronecker product formalism to rewrite the gradient $\nabla_{\mathbf{F}}\mathcal{E}_\theta(\mathbf{F})$ as a vector in $\mathbb{R}^{nd}$: explicitly, we get

$$\frac{1}{2}\nabla_{\mathbf{F}}\mathcal{E}_\theta(\mathbf{F}) = (\mathbf{\Omega}\otimes\mathbf{I}_n - \mathbf{W}\otimes\mathbf{A})\mathrm{vec}(\mathbf{F}) + (\tilde{\mathbf{W}}\otimes\mathbf{I}_n)\mathrm{vec}(\mathbf{F}(0)).$$

It follows then that

$$\frac{1}{2}\frac{d\mathcal{E}_\theta(\mathbf{F}(t))}{dt} = \frac{1}{2}\left(\nabla_{\mathbf{F}}\mathcal{E}_\theta(\mathbf{F}(t))\right)^\top\mathrm{vec}(\dot{\mathbf{F}}(t)) =$$
$$= \left(\frac{1}{2}\nabla_{\mathbf{F}}\mathcal{E}_\theta(\mathbf{F}(t))\right)^\top\sigma\left(-\frac{1}{2}\nabla_{\mathbf{F}}\mathcal{E}_\theta(\mathbf{F}(t))\right).$$

If we introduce the notation $\mathbf{Z}(t) = -\frac{1}{2}\nabla_{\mathbf{F}}\mathcal{E}_\theta(\mathbf{F}(t))$, then we can rewrite the derivative as

$$\frac{d\mathcal{E}_\theta(\mathbf{F}(t))}{dt} = -\mathbf{Z}(t)^\top\sigma(\mathbf{Z}(t)) = -\sum_\alpha \mathbf{Z}(t)^\alpha\sigma(\mathbf{Z}(t)^\alpha) \leq 0$$

by assumption on $\sigma$. The discrete case follows similarly. Let us use the same notation as above so we can write $\mathbf{F}(t+\tau) = \mathbf{F}(t) + \tau\sigma(\mathbf{Z}(t))$, with $\mathbf{Z}(t) = -\frac{1}{2}\nabla_{\mathbf{F}}\mathcal{E}_\theta(\mathbf{F}(t))$.

$$\begin{aligned}
\mathcal{E}_\theta(\mathbf{F}(t+\tau)) &= \langle\mathrm{vec}(\mathbf{F}(t+\tau)), (\mathbf{\Omega}\otimes\mathbf{I}_n - \mathbf{W}\otimes\mathbf{A})\mathrm{vec}(\mathbf{F}(t+\tau)) + 2(\tilde{\mathbf{W}}\otimes\mathbf{I}_n)\mathrm{vec}(\mathbf{F}(0))\rangle \\
&= \langle\mathrm{vec}(\mathbf{F}(t)) + \tau\sigma(\mathbf{Z}(t)), (\mathbf{\Omega}\otimes\mathbf{I}_n - \mathbf{W}\otimes\mathbf{A})\mathrm{vec}(\mathbf{F}(t+\tau)) + 2(\tilde{\mathbf{W}}\otimes\mathbf{I}_n)\mathrm{vec}(\mathbf{F}(0))\rangle \\
&= \langle\mathrm{vec}(\mathbf{F}(t)) + \tau\sigma(\mathbf{Z}(t)), (\mathbf{\Omega}\otimes\mathbf{I}_n - \mathbf{W}\otimes\mathbf{A})\left(\mathrm{vec}(\mathbf{F}(t) + \tau\sigma(\mathbf{Z}(t))) + 2(\tilde{\mathbf{W}}\otimes\mathbf{I}_n)\mathrm{vec}(\mathbf{F}(0))\right)\rangle \\
&= \langle\mathrm{vec}(\mathbf{F}(t)), (\mathbf{\Omega}\otimes\mathbf{I}_n - \mathbf{W}\otimes\mathbf{A})\mathrm{vec}(\mathbf{F}(t)) + 2(\tilde{\mathbf{W}}\otimes\mathbf{I}_n)\mathrm{vec}(\mathbf{F}(0))\rangle \\
&\quad + \tau\langle\mathrm{vec}(\mathbf{F}(t)), (\mathbf{\Omega}\otimes\mathbf{I}_n - \mathbf{W}\otimes\mathbf{A})\sigma(\mathbf{Z}(t))\rangle \\
&\quad + \tau\langle\sigma(\mathbf{Z}(t)), (\mathbf{\Omega}\otimes\mathbf{I}_n - \mathbf{W}\otimes\mathbf{A})\mathrm{vec}(\mathbf{F}(t) + 2(\tilde{\mathbf{W}}\otimes\mathbf{I}_n)\mathrm{vec}(\mathbf{F}(0))\rangle \\
&\quad + \tau^2\langle\sigma(\mathbf{Z}(t)), (\mathbf{\Omega}\otimes\mathbf{I}_n - \mathbf{W}\otimes\mathbf{A})\sigma(\mathbf{Z}(t))\rangle.
\end{aligned}$$

By using that $\mathbf{\Omega}\otimes\mathbf{I}_n - \mathbf{W}\otimes\mathbf{A}$ is symmetric, we find that

$$\begin{aligned}
\mathcal{E}_\theta(\mathbf{F}(t+\tau)) &= \mathcal{E}_\theta(\mathbf{F}(t)) + 2\tau\langle\sigma(\mathbf{Z}(t), (\mathbf{\Omega}\otimes\mathbf{I}_n - \mathbf{W}\otimes\mathbf{A})\mathrm{vec}(\mathbf{F}(t) + (\tilde{\mathbf{W}}\otimes\mathbf{I}_n)\mathrm{vec}(\mathbf{F}(0)))\rangle \\
&\quad + \tau^2\langle\frac{1}{\tau}(\mathbf{F}(t+\tau) - \mathbf{F}(t)), (\mathbf{\Omega}\otimes\mathbf{I}_n - \mathbf{W}\otimes\mathbf{A})\frac{1}{\tau}(\mathbf{F}(t+\tau) - \mathbf{F}(t))\rangle \\
&= \mathcal{E}_\theta(\mathbf{F}(t)) - 2\tau\langle\sigma(\mathbf{Z}(t)), \mathbf{Z}(t)\rangle + \langle\mathbf{F}(t+\tau) - \mathbf{F}(t), (\mathbf{\Omega}\otimes\mathbf{I}_n - \mathbf{W}\otimes\mathbf{A})(\mathbf{F}(t+\tau) - \mathbf{F}(t))\rangle \\
&\leq \mathcal{E}_\theta(\mathbf{F}(t)) + C_+\|\mathbf{F}(t+\tau) - \mathbf{F}(t))\|^2,
\end{aligned}$$

where again we have used that $\mathbf{Z}^\top\sigma(\mathbf{Z}) \geq 0$. This completes the proof. $\qquad\square$

To further support the principle that the effects induced by $\mathbf{W}$ are similar even in this non-linear setting, we consider a simplified scenario.

**Lemma D.1.** *If we choose $\mathbf{\Omega} = \mathbf{W} = \mathrm{diag}(\boldsymbol{\omega})$ with $\omega^r \leq 0$ for $0 \leq r \leq d-1$ and $\tilde{\mathbf{W}} = \mathbf{0}$ i.e. $t \mapsto \mathbf{F}(t)$ solves the dynamical system*

$$\dot{\mathbf{F}}(t) = \sigma\left(-\mathbf{\Delta}\mathbf{F}(t)\mathrm{diag}(\boldsymbol{\omega})\right),$$

*with $x\sigma(x) \geq 0$, then the standard graph Dirichlet energy satisfies*

$$\frac{d\mathcal{E}^{\mathrm{Dir}}(\mathbf{F}(t))}{dt} \geq 0.$$

*Proof.* This again simply follows from directly computing the derivative:

$$\frac{d\mathcal{E}^{\mathrm{Dir}}(\mathbf{F}(t))}{dt} = \frac{1}{2}\frac{d}{dt}\Big(\sum_{r=1}^{d}\sum_{(i,j)\in\mathsf{E}}\Big(\frac{f_i^r(t)}{\sqrt{d_i}} - \frac{f_j^r(t)}{\sqrt{d_j}}\Big)^2\Big)$$

$$= \sum_{r=1}^{d}\sum_{i\in\mathsf{V}}(\mathbf{\Delta f}^r)_i\sigma(-\omega^r(\mathbf{\Delta f}^r)_i) = \sum_{r=1}^{d}\sum_{i\in\mathsf{V}}(\mathbf{\Delta f}^r)_i\sigma(|\omega^r|(\mathbf{\Delta f}^r)_i) \geq 0.$$

**Important consequence:** The previous Lemma implies that even with non-linear activations, negative eigenvalues of the channel-mixing induce repulsion and indeed the solution becomes less smooth as measured by the classical Dirichlet Energy increasing along the solution. Generalising this result to more arbitrary choices is not immediate and we reserve this for future work.

$\square$

# E    Additional details on experiments

## E.1    Additional results on real-world tasks

**General details.**    The graph-convolutional models in (16) and (17) are implemented in PyTorch (Paszke et al., 2019), using PyTorch geometric Fey & Lenssen (2019) and torchdiffeq (Chen et al., 2018). Hyperparameters were tuned using wandb (Biewald, 2020) and random grid search. Experiments were run on AWS p2.8xlarge machines, each with 8 Tesla V100-SXM2 GPUs.

We further improve our results on the node-classification tasks of by considering the following, more general, parameterizations. We consider an instance of the gradient-flow framework, termed GRAFF, of the form:

$$\text{GRAFF}: \quad \mathbf{F}(t+\tau) = \mathbf{F}(t) + \tau\left(-\mathbf{F}(t)\mathrm{diag}(\boldsymbol{\omega}) + \mathbf{AF}(t)\mathbf{W} - \beta\mathbf{F}(0)\right), \tag{34}$$

where $\boldsymbol{\omega}\in\mathbb{R}^d$ and $\beta\in\mathbb{R}$, $\mathbf{W}$ is a *symmetric* $d\times d$-matrix *shared* across layers and (node-wise) encoder and decoder are MLPs. We descibe different implementations of $\mathbf{W}$ below. In accordance with Theorem 6.1, we also consider a non-linear variant (termed GRAFF$_{\mathrm{NL}}$) as

$$\text{GRAFF}_{\mathrm{NL}}: \quad \mathbf{F}(t+\tau) = \mathbf{F}(t) + \tau\sigma\left(-\mathbf{F}(t)\mathrm{diag}(\boldsymbol{\omega}) + \mathbf{AF}(t)\mathbf{W} - \beta\mathbf{F}(0)\right), \tag{35}$$

with $\sigma$ s.t. $x\sigma(x) \geq 0$. We note that differently from (16) and (17), these models also allow for a source term controlled by $\beta$ and a diagonal term acting on the state of a node, independent of the graph structure, controlled by $\boldsymbol{\omega}$.

**Methodology.**    In the experiments we rely on the following parameterizations. We implement encoder $\psi_{\mathrm{EN}}$ and $\psi_{\mathrm{DE}}$ as single linear layers or MLPs. We consider *diagonally-dominant* (DD) and *diagonal* (D) choices for the structure of $\mathbf{W}$ that offer explicit control over its spectrum. In the (DD)-case, we consider a $\mathbf{W}^0\in\mathbb{R}^{d\times d}$ *symmetric* with zero diagonal and $\mathbf{w}\in\mathbb{R}^d$ defined by $\mathbf{w}_\alpha = q_\alpha\sum_\beta|\mathbf{W}^0_{\alpha\beta}| + r_\alpha$, and set $\mathbf{W} = \mathrm{diag}(\mathbf{w}) + \mathbf{W}^0$. Due to the Gershgorin Theorem the eigenvalues of $\mathbf{W}$ belong to $[\mathbf{w}_\alpha - \sum_\beta|\mathbf{W}^0_{\alpha\beta}|, \mathbf{w}_\alpha + \sum_\beta|\mathbf{W}^0_{\alpha\beta}|]$, so the model 'can' easily re-distribute mass in the spectrum of $\mathbf{W}$ via $q_\alpha, r_\alpha$. This generalizes the decomposition of $\mathbf{W}$ in Chen et al. (2020) providing a justification in terms of its spectrum. For (D) we take $\mathbf{W}$ to be diagonal.

**Real world experiments.** In Table 3 we test GRAFF and GRAFF$_{\mathrm{NL}}$ on the same datasets with varying homophily adopted in Section 7 (Sen et al., 2008; Rozemberczki et al., 2021; Pei et al., 2020). We use results provided in Yan et al. (2021, Table 1), which include GCNs models, GAT (Veličković et al., 2018), PairNorm (Zhao & Akoglu, 2019) and models designed for heterophily (GGCN (Yan et al., 2021), Geom-GCN (Pei et al., 2020), H2GCN (Zhu et al., 2020) and GPRGNN (Chien et al., 2021)). For Sheaf (Bodnar et al., 2022), a recent strong baseline with heterophily, we took the best performing variant (out of six) for each dataset. We include continuous baselines CGNN (Xhonneux et al., 2020) and GRAND (Chamberlain et al., 2021a) to corroborate Theorem 5.4. Training, validation and test splits are taken from Pei et al. (2020) for all datasets

|  | Texas | Wisconsin | Cornell | Film | Squirrel | Chameleon | Citeseer | Pubmed | Cora |
|---|---|---|---|---|---|---|---|---|---|
| Hom level | **0.11** | **0.21** | **0.30** | **0.22** | **0.22** | **0.23** | **0.74** | **0.80** | **0.81** |
| #Nodes | 183 | 251 | 183 | 7,600 | 5,201 | 2,277 | 3,327 | 18,717 | 2,708 |
| #Edges | 295 | 466 | 280 | 26,752 | 198,493 | 31,421 | 4,676 | 44,327 | 5,278 |
| #Classes | 5 | 5 | 5 | 5 | 5 | 5 | 7 | 3 | 6 |
| GGCN | $84.86 \pm 4.55$ | $86.86 \pm 3.29$ | $85.68 \pm 6.63$ | $37.54 \pm 1.56$ | $55.17 \pm 1.58$ | $71.14 \pm 1.84$ | $77.14 \pm 1.45$ | $89.15 \pm 0.37$ | $87.95 \pm 1.05$ |
| GPRGNN | $78.38 \pm 4.36$ | $82.94 \pm 4.21$ | $80.27 \pm 8.11$ | $34.63 \pm 1.22$ | $31.61 \pm 1.24$ | $46.58 \pm 1.71$ | $77.13 \pm 1.67$ | $87.54 \pm 0.38$ | $87.95 \pm 1.18$ |
| H2GCN | $84.86 \pm 7.23$ | $87.65 \pm 4.98$ | $82.70 \pm 5.28$ | $35.70 \pm 1.00$ | $36.48 \pm 1.86$ | $60.11 \pm 2.15$ | $77.11 \pm 1.57$ | $89.49 \pm 0.38$ | $87.87 \pm 1.20$ |
| GCNII | $77.57 \pm 3.83$ | $80.39 \pm 3.40$ | $77.86 \pm 3.79$ | $37.44 \pm 1.30$ | $38.47 \pm 1.58$ | $63.86 \pm 3.04$ | $77.33 \pm 1.48$ | $90.15 \pm 0.43$ | $88.37 \pm 1.25$ |
| Geom − GCN | $66.76 \pm 2.72$ | $64.51 \pm 3.66$ | $60.54 \pm 3.67$ | $31.59 \pm 1.15$ | $38.15 \pm 0.92$ | $60.00 \pm 2.81$ | $78.02 \pm 1.15$ | $89.95 \pm 0.47$ | $85.35 \pm 1.57$ |
| PairNorm | $60.27 \pm 4.34$ | $48.43 \pm 6.14$ | $58.92 \pm 3.15$ | $27.40 \pm 1.24$ | $50.44 \pm 2.04$ | $62.74 \pm 2.82$ | $73.59 \pm 1.47$ | $87.53 \pm 0.44$ | $85.79 \pm 1.01$ |
| GraphSAGE | $82.43 \pm 6.14$ | $81.18 \pm 5.56$ | $75.95 \pm 5.01$ | $34.23 \pm 0.99$ | $41.61 \pm 0.74$ | $58.73 \pm 1.68$ | $76.04 \pm 1.30$ | $88.45 \pm 0.50$ | $86.90 \pm 1.04$ |
| GCN | $55.14 \pm 5.16$ | $51.76 \pm 3.06$ | $60.54 \pm 5.30$ | $27.32 \pm 1.10$ | $53.43 \pm 2.01$ | $64.82 \pm 2.24$ | $76.50 \pm 1.36$ | $88.42 \pm 0.50$ | $86.98 \pm 1.27$ |
| GAT | $52.16 \pm 6.63$ | $49.41 \pm 4.09$ | $61.89 \pm 5.05$ | $27.44 \pm 0.89$ | $40.72 \pm 1.55$ | $60.26 \pm 2.50$ | $76.55 \pm 1.23$ | $87.30 \pm 1.10$ | $86.33 \pm 0.48$ |
| MLP | $80.81 \pm 4.75$ | $85.29 \pm 3.31$ | $81.89 \pm 6.40$ | $36.53 \pm 0.70$ | $28.77 \pm 1.56$ | $46.21 \pm 2.99$ | $74.02 \pm 1.90$ | $75.69 \pm 2.00$ | $87.16 \pm 0.37$ |
| CGNN | $71.35 \pm 4.05$ | $74.31 \pm 7.26$ | $66.22 \pm 7.69$ | $35.95 \pm 0.86$ | $29.24 \pm 1.09$ | $46.89 \pm 1.66$ | $76.91 \pm 1.81$ | $87.70 \pm 0.49$ | $87.10 \pm 1.35$ |
| GRAND | $75.68 \pm 7.25$ | $79.41 \pm 3.64$ | $82.16 \pm 7.09$ | $35.62 \pm 1.01$ | $40.05 \pm 1.50$ | $54.67 \pm 2.54$ | $76.46 \pm 1.77$ | $89.02 \pm 0.51$ | $87.36 \pm 0.96$ |
| Sheaf(max) | $85.95 \pm 5.51$ | $89.41 \pm 4.74$ | $84.86 \pm 4.71$ | $37.81 \pm 1.15$ | $56.34 \pm 1.32$ | $68.04 \pm 1.58$ | $76.70 \pm 1.57$ | $89.49 \pm 0.40$ | $86.90 \pm 1.13$ |
| GRAFF | $88.38 \pm 4.53$ | $88.83 \pm 3.29$ | $84.05 \pm 6.10$ | $37.11 \pm 1.08$ | $58.72 \pm 0.84$ | $71.08 \pm 1.75$ | $77.30 \pm 1.85$ | $90.04 \pm 0.41$ | $88.01 \pm 1.03$ |
| GRAFF$_{\text{NL}}$ | $86.49 \pm 4.84$ | $87.26 \pm 2.52$ | $77.30 \pm 3.24$ | $35.96 \pm 0.95$ | $59.01 \pm 1.31$ | $71.38 \pm 1.47$ | $76.81 \pm 1.12$ | $89.81 \pm 0.50$ | $87.81 \pm 1.13$ |

Table 3: Node-classification results. Top three models are coloured by First, Second, Third

| dataset | Texas | Wisconsin | Cornell | Film | Squirrel | Chameleon | Citeseer | Pubmed | Cora |
|---|---|---|---|---|---|---|---|---|---|
| Homophily | 0.11 | 0.21 | 0.3 | 0.22 | 0.22 | 0.23 | 0.74 | 0.8 | 0.81 |
| Depth | 2 | 2 | 2 | 2 | 4 | 4 | 2 | 2 | 4 |
| $\mu_{d-1}$ | 2.07 | 1.33 | 1.37 | 3.33 | 2.48 | 2.66 | 4.1 | 9.47 | 2.49 |
| $\mu_0$ | -3.07 | -0.82 | -0.76 | -1.45 | -3.61 | -3.72 | -0.96 | -3.89 | -0.26 |
| mean($\{\mu_r\}$) | -0.02 | 0.02 | 0.02 | 0.06 | -0.1 | -0.1 | 0.13 | 0.15 | 0.11 |
| $\mathcal{E}^{\text{Dir}}(\mathbf{F}(0))$ | 13171 | 5367 | 3687 | 7460 | 68811 | 25772 | 4749 | 5737 | 1226 |
| $\mathcal{E}^{\text{Dir}}(\mathbf{F}(m))$ | 42674 | 6767 | 4932 | 12224 | 828532 | 398302 | 37168 | 119518 | 33700 |
| $\mathcal{E}^{\text{Dir}}(\mathbf{F}(0))/\|\mathbf{F}(0)\|^2$ | 1.68 | 1.33 | 1.35 | 1.1 | 1.77 | 1.86 | 0.55 | 0.82 | 0.57 |
| $\mathcal{E}^{\text{Dir}}(\mathbf{F}(m))/\|\mathbf{F}(m)\|^2$ | 1.67 | 0.71 | 0.91 | 0.71 | 1.47 | 1.21 | 0.16 | 0.29 | 0.11 |
| Test accuracy (mean) | 81.08 | 81.96 | 72.43 | 34.16 | 52.48 | 68.99 | 76.43 | 88.56 | 87.4 |
| Test accuracy (std) | 4.52 | 4.45 | 7.13 | 0.9 | 1.8 | 1.7 | 1.53 | 0.51 | 0.97 |

Table 4: SGCN$_{gf}$ learned spectrum

for comparison. For the Cornell dataset we take the adjacency file from before the July 2022 update of Pei et al. (2020).

**Results.** GRAFF and GRAFF$_{\text{NL}}$ are both *versions of graph convolutions with stronger 'inductive bias' given by the energy $\mathcal{E}_\theta$ decreasing along the solution*; in fact, we can recover them from graph convolutions by simply requiring that the channel-mixing is *symmetric and shared across layers*. Nonetheless they achieve competitive results on all datasets often outperforming slower and more complex models. As noted in Section 7, the improved performance is mainly due to the presence of a residual connection at each layer, which simply enables the channel-mixing matrices to induce either attraction or repulsion, via their eigenvalues.

A word of caution though: these tasks are very sensitive to the hyperparameter tuning, making them generally not ideal candidates to probe different models. While they suffice for our purposes, given the theoretical nature of our work, we believe them to be at times very poor indicators of the 'heterophilic problem'.

## E.2 Details on hyperparameters

Using wandb Biewald (2020) we performed a random grid search with uniform sampling of the continuous variables. We provide the hyperparameters that achieved the best results from the random grid search in Table 5. Input dropout and dropout are the rates applied to the encoder/decoder respectively *with no dropout applied in the ODE block*. Further hyperparameters decide the use of non-linearities, batch normalisation, parameter vector $\omega$ and source term multiplier $\beta$ which are specified in the code.

|           | w_style  | lr     | decay  | dropout | input_dropout | hidden | time | step_size |
|-----------|----------|--------|--------|---------|---------------|--------|------|-----------|
| chameleon | diag_dom | 0.0050 | 0.0005 | 0.36    | 0.48          | 64     | 3.33 | 1         |
| squirrel  | diag_dom | 0.0065 | 0.0009 | 0.17    | 0.35          | 128    | 2.87 | 1         |
| texas     | diag_dom | 0.0041 | 0.0354 | 0.33    | 0.39          | 64     | 0.6  | 0.5       |
| wisconsin | diag     | 0.0029 | 0.0318 | 0.37    | 0.37          | 64     | 2.1  | 0.5       |
| cornell   | diag     | 0.0021 | 0.0184 | 0.30    | 0.44          | 64     | 2.0  | 1         |
| film      | diag     | 0.0026 | 0.0130 | 0.48    | 0.42          | 64     | 1.5  | 1         |
| Cora      | diag     | 0.0026 | 0.0413 | 0.34    | 0.53          | 64     | 3.0  | 0.25      |
| Citeseer  | diag     | 0.0001 | 0.0274 | 0.22    | 0.51          | 64     | 2.0  | 0.5       |
| Pubmed    | diag     | 0.0039 | 0.0003 | 0.42    | 0.41          | 64     | 2.6  | 0.5       |

Table 5: Selected hyperparameters for real-world datasets

## E.3 Additional details on the spectral experiments

We report additional details on the spectrum of $\mathbf{W}_S$ in (16) and on the normalized Dirichlet energy in Table 4. We also report the distribution of the eigenvalues of the learned channel-mixing matrix $\mathbf{W}_S$ in Figure 4.

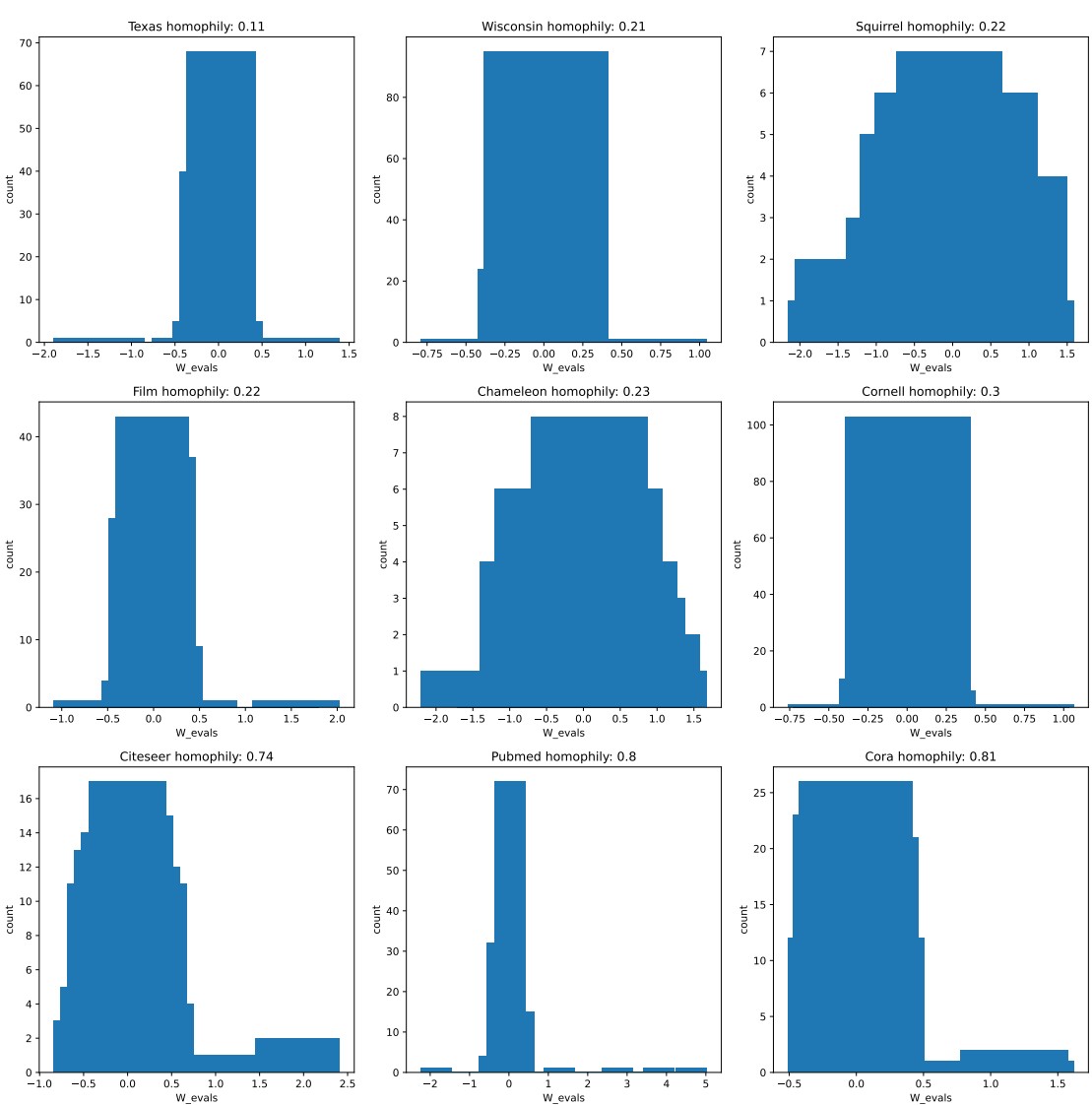

Figure 4: Histogram of $\mathsf{SGCN}_{gf}$ learned spectrum

