# OpenReview forum: "Understanding convolution on graphs via energies"
_TMLR — Accepted by TMLR_

### Review · Reviewer_fVAu · 2023-07-06

**Summary Of Contributions:**

This paper analyzes the behavior of node features of a GNN represented by a gradient flow or its discretized version. Continuous-time GNN is defined as a gradient flow by parametric energy determined by node features (Eq. (6)). Then, this paper introduces the concepts of low-frequency-dominant (LFD) and high-frequency-dominant (HFD), which quantitatively evaluate the smoothness of the node representation using the normalized energy. This paper shows that a GNN can be both LFD and HFD depending on the spectra of weight matrices. Furthermore, this paper introduces a residual GNN as a discretized version of the continuous-time GNN and shows similar results. Also, it is shown that the discretized GNN is always LFD when there is no residual connection. The proposed model was applied to the node classification task on various heterophily datasets to verify its consistency with the theoretical analysis.

**Audience:**

Yes

**Broader Impact Concerns:**

There are no major concerns about broader impacts.

**Claims And Evidence:**

Yes

**Requested Changes:**

P.5: As explained at the beginning of Section 4, the main characteristic of gradient flow is that the energy decreases monotonically along the solution. However, this paper does not use such a property of gradient flow. Instead, the normalized energy converges to the maximum possible value under certain conditions in Theorem 4.3. I wonder how the theoretical analysis utilizes the benefit of gradient flows, such as the monotonically decreasing property.

P.6: The *only if* part of Proposition 4.1 does not hold mathematically strictly speaking (i.e., if Equation (8) is the gradient flow of energy Equation (6), then, $\boldsymbol{\Omega}$ and $\boldsymbol{\mathrm{W}}$ are symmetric.) For example, in the extreme case where the graph has no edges (i.e., $\boldsymbol{\mathsf{A}}=0$), no constraint is imposed on $\boldsymbol{\mathrm{W}}$, which does not have to be symmetric in particular. Even if the only-if part of this proposition is not satisfied, it does not cause critical problems in constructing the proposed model. However, a more elaborate discussion is needed to make it a mathematically correct statement.

P.8:
> [...] (ii) graph convolutions can avoid over-smoothing through the negative eigenvalues of the channel-mixing by incurring the opposite behaviour of over-shaping [...]

This sentence implicitly assumes that $\boldsymbol{\mathrm{W}}$ has at least one negative eigenvalue when (12) holds. It is certainly true (if $\mu_0\geq 0$, then $\lambda_{n-1}\leq 2$ implies $\mu_{d-1}/(\lambda_{n-1}-1)\geq \mu_{d-1} \geq \mu_0$, which contradicts to (12).) However since it is not obvious, I suggest writing the proof explicitly.

P.10, Theorem 6.1: [...] with $\boldsymbol{\Omega}, \boldsymbol{W}$ symmetric, [...] -> [...] with $\boldsymbol{\Omega}$ and $\boldsymbol{W}$ being symmetric, [...]

P.10, Theorem 6.1: [...] let $c$ denote the most positive eigenvalue of $\boldsymbol{\Omega} \otimes \boldsymbol{I}_n - \boldsymbol{W} \otimes \boldsymbol{\mathsf{A}}$ [...]: Can we assume that there always exists positive eigenvalues?

P.11:
> We emphasize that this is in support of our theoretical findings, since we proved that thanks to a residual connection, graph-convolutional models enable the channel-mixing to also generate repulsion along the edges through its negative eigenvalues hence compensating for the underlying heterophily of the graphs.

This sentence implicitly assumes that the learned weights have negative eigenvalues. However, I wonder whether it is empirically confirmed in numerical experiments.

P.11: Table 2: I think it is easier to read if we reverse the denominator and numerator of $\frac{\mathcal{E}^{\mathrm{Dir}}(\boldsymbol{\mathrm{F}}(0))/\|\boldsymbol{\mathrm{F}}(0)\|^2}{\mathcal{E}^{\mathrm{Dir}}(\boldsymbol{\mathrm{F}}(m))/\|\boldsymbol{\mathrm{F}}(m)\|^2}$

P.12: enhanche -> enhance

**Strengths And Weaknesses:**

**Strengths**

- Based on normalized Dirichlet energy, over-smoothing and its opposite state (called over-shaping in this paper) are defined in a unified manner (Definition 4.2.)
- In the previous analysis of over-smoothing, only the scale of the GNN (specifically, the singular value) was considered. This paper analyzes more precisely the role of the weight matrix and finds that the negative eigenvalue is helpful for avoiding over-smoothing (Theorem 4.3)
- The effect of residual connection on over-smoothing is further clarified (Theorem 5.3).


**Weaknesses**

- This paper first introduces the general form of the model as Equations (1) or (8). Equation (1) claims its usefulness because it includes GCN, GraphSAGE, GCNII, and GIN with residual connections as special cases. However, only its special cases are considered in most analyses. Specifically, Section 4.2.2 considers the case $\boldsymbol{\mathrm{\tilde{W}}}_t=0$ and $\boldsymbol{\mathsf{A}}=D^{-1/2}AD^{-1/2}$. Also, Section 4.3, Theorem 5.1, and Corollary 5.2 further impose the constraint $\boldsymbol{\Omega}_t=0$. Among the above examples above, only GCN satisfies this parametrization. The exception is Proposition 4.1, which used the general form of (8). However, since it only provides the necessary conditions for defining a GNN as a gradient flow, I think it may not answer why we should consider the ODE of the form (1) or (8).
- The energy $\mathcal{E}_\theta$ is the Diriclet energy when $\boldsymbol{\mathrm{\Omega}}=\boldsymbol{\mathrm{W}}=\boldsymbol{\mathrm{I}}$ (and $\boldsymbol{\tilde{\mathrm{W}}}=0$) (Equation (5)). Since the model analyzed in Theorem 4.3 assumes $\boldsymbol{\mathrm{\Omega}}=\boldsymbol{\mathrm{W}}=0$, the model is not a gradient flow of Dirichlet energy (or at least, it is not shown explicitly). However, LFD and HFD are defined using normalized Dirichlet energy. So, the model and the LFD/HFD definitions use different energies. This fact raises questions about how the theoretical analysis (e.g., Theorem 4.3) uses the fact that the proposed model is defined as a gradient flow with $\mathcal{E}_\theta$.

---

> ### Author Response · Authors · 2023-07-07
> **Thanks for the review, response**
>
> Thanks for the review and the feedback. We have provided a list of the general updates to the manuscript above, while here we address all your points in order.
>
> “ _This paper first introduces the general form of the model as Equations (1) or (8). (...)_ ”
>
> Thanks for the feedback. Eq. (1) is used in its generality on two occasions: (i) proving when its linear version is the gradient flow of an energy, and (ii) deriving Theorem 6.1, which pertains to the analysis of the monotonicity of non-linear layers. The simplified (GCN-like) versions of Eq. 1 are instead used for the derivations of the exact LFD/HFD behaviours for both the discrete and continuous cases (Theorem 4.1 and Theorem 5.1).
>
> We think that starting from a general equation is helpful, since it highlights when this equation can be casted as gradient flow (in fact Proposition 4.1 gives necessary and sufficient conditions, see below for more details), and we further prove that even in the non-linear case we can derive monotonicity properties. However, we fully agree that some further clarification and separation could help the presentation. In fact, to answer Q.1 and Q.2 in the introduction, it suffices to show that _some_ graph convolutions can already avoid low-pass filter behaviours and over-smoothing, which we do by showing that GCN-type of MPNNs may enhance the high frequencies and induce over-sharpening (HFD behaviour).
>
> In this regard, we have **modified the manuscript as follows** (we copy here the relevant part in the general comments above):
>
> - At page 7 above the statement of Theorem 4.3, we have added a paragraph “Similarly to..” where we clarify that the exact spectral analysis in the time continuous and time discrete cases is performed for a simplified version of the gradient flow system. We have also stated more visibly the assumption used throughout the rest of Section 4 and Section 5 to help the reader. We note that the results can be extended to versions of the more general gradient flow equations of Proposition 1 quite trivially (e.g. when $\boldsymbol{\Omega}, \mathbf{W}$ commute), however this could deviate too much from our story. In fact, the purpose of our work is showing that there exist simple graph convolutions that can enhance the high frequencies and induce behaviours other than over-smoothing. On the other hand, we highlight that the results in Section 6 for the non-linear layers hold in the generality of Eq. 1.
>
> - In line with the previous point, we have modified Eq. (10) at the beginning of Section 5 to be the simplified gradient flow system we consider for Theorem 5.1.
>
> “ _The energy $\mathcal{E}\_\theta$ is the Dirichlet energy when (...)._ ”
>
>
> We think that there might be some misunderstanding here that we hope to clarify. It is certainly true that the model we study is the gradient flow of an energy that, in general, does _not_ coincide with the Dirichlet energy. This is **actually needed**, since if the model was the gradient flow of the Dirichlet energy, then we would necessarily have the heat equation. The idea of introducing LFD/HFD dynamics based on the Dirichlet energy is still well-posed though. For example, for _any_ dynamical system (continuous or discrete in time) one can still ask if the Dirichlet energy is increasing or decreasing along the features and in fact this approach has been used to formally characterize the over-smoothing phenomenon. We follow a similar approach, and ask if the Dirichlet energy is increasing or decreasing along our gradient flow equations. Since our equations minimize an energy _other_ than Dirichlet, it turns out that under specific conditions the Dirichlet energy may actually increase (maximally) along the features. In turn, this entails that over-smoothing is avoided and over-sharpening actually arises. The idea behind using the Dirichlet energy to study the dynamics of a given gradient flow is that the Dirichlet energy is associated with the smoothness of the features and can hence be used as a measurement for whether the given system of equations has smoothing or sharpening effects.

---

> ### Author Response · Authors · 2023-07-07
> **Response part II**
>
> We hope this clarifies this important point.
>
> “ _As explained at the beginning of Section 4, the main characteristic of gradient flow is that the energy decreases monotonically along the solution (...)_ "
>
>
> Thank you for your point. We think that here there is a similar misunderstanding as in the comment above. Consider the statement of Theorem 4.3 or equivalently the discretized version in Theorem 5.1. For both cases, the system is a gradient flow of an energy $\mathcal{E}\_\theta$ _different_ than the Dirichlet energy, which is decreasing as time (depth) increases. It is precisely because $\mathcal{E}\_\theta$ is more general than the Dirichlet energy that even though $\mathcal{E}\_\theta$ is decreasing, it can still promote repulsion along edges. What we are doing here is simply using the normalized Dirichlet energy as a yardstick for measuring if the gradient flow is increasing or reducing the smoothness of the features. What we show in Theorem 4.3 and Theorem 5.1 is that for certain weights, the energy associated with the gradient flow reduces the smoothness of the features, a property that can be measured by _the normalized Dirichlet energy_ attaining its maximum. We hope this clarifies the confusion and we are happy to comment on any standing doubt.
>
> “ _The only if part of Proposition 4.1 does not hold mathematically strictly speaking (...)_ ”
>
> Thanks for checking our statements carefully. The only if Proposition actually holds as long as the graph has a non-trivial edge. We have added this extra requirement to avoid corner cases and formally expanded the proof of the Proposition in Appendix B.1, pag. 20.
>
> “ _This sentence implicitly assumes that $\mathbf{W}$  has at least one negative eigenvalue_ ”
>
> Thanks for the detailed note. It actually follows from (12) that $\mathbf{W}$ must have a negative eigenvalue. Explicitly, since we are ordering the eigenvalues of $\mathbf{W}$ from lowest to largest, if the first inequality of (12) holds, then $\mu_0 < 0$, otherwise, if it was positive, then we could remove the absolute value and derive that $\mu_0(\lambda_{n-1} -1) > \mu_{d-1}$, but now $\mu_{d-1}\geq \mu_0$ and $\lambda_{n-1} - 1 \leq 1$, which is a contradiction. To clarify this point, we have added a sentence below Eq. (12) mentioning that when (12) is satisfied, then $\mu_0$ must be negative “Note that (...)”.
>
> “ _With $\boldsymbol{\Omega}, \mathbf{W}$ being symmetric_ ”
>
> Thanks for the pointer, we have corrected the typo accordingly.
>
> " _P.10, Theorem 6.1: (...)  Can we assume that there always exists positive eigenvalues?_ "
>
> Thanks for reading the statements carefully. The sentence means that if there are no positive eigenvalues, then we can simply take c to be zero. We have added this clarification in the statement of Theorem 6.1.
>
> “ _This sentence implicitly assumes that the learned weights have negative eigenvalues. However, I wonder whether it is empirically confirmed in numerical experiments._ ”
>
> Thanks for the comment. We have actually confirmed this point empirically in Table 2, where you can see that the weight matrices learn negative eigenvalues: since $\mu_0 < 0$ and $\mu_{d-1} > 0$ the quantity $-\mu_{d-1}/\mu_{0}$ is positive, and the smaller this number the larger the absolute value of the most negative eigenvalue compared to the most positive one. As you can see, there seems to be a correlation between the underlying heterophily of the graph and the required magnitude of the negative eigenvalues.
>
>
> “ _Table 2: I think it is easier to read if we reverse the denominator and numerator_ ”
>
> We have reversed the numerator and denominator as suggested.
>
> “_ Enhanche_ ”
>
> Typo corrected, thanks for reading the manuscript carefully!

---

> ### Author Response · Authors · 2023-07-21
> **End of rebuttal**
>
> We just wanted to point out that the rebuttal period is ending, so we hope we have clarified your doubts and that the modifications to the manuscript we made, partly based on your feedback, helped improve the submission.
>
> We also received a notification from open review that seems to suggest that you edited your review, however we were not able to identify how or where?

---

> > ### Comment · Reviewer_fVAu · 2023-08-05
> > **Response to authors' comments.**
> >
> > Thank you for your responses, and I am sorry that I have not responded to them. Your responses clarified my concerns, and I have no further questions.
> >
> > > We also received a notification from open review that seems to suggest that you edited your review, however we were not able to identify how or where?
> >
> > I fixed a small typo in the following sentence (I removed Therefore). I am sorry for confusing you
> >
> > Since the model analyzed in Theorem 4.3 assumes $\boldsymbol{\mathrm{\Omega}}=\boldsymbol{\mathrm{W}}=0$. Therefore, the model is not a gradient flow of Dirichlet energy (or at least, it is not shown explicitly). However, LFD and HFD are defined using normalized Dirichlet energy.
> > ->
> > Since the model analyzed in Theorem 4.3 assumes $\boldsymbol{\mathrm{\Omega}}=\boldsymbol{\mathrm{W}}=0$, the model is not a gradient flow of Dirichlet energy (or at least, it is not shown explicitly). However, LFD and HFD are defined using normalized Dirichlet energy.

---

> > > ### Author Response · Authors · 2023-08-05
> > > **Thanks**
> > >
> > > The model analyzed in Theorem 4.3 is not the gradient flow of the Dirichlet Energy---that would happen only if $\boldsymbol{\Omega} = \mathbf{W} = \mathbf{I}$, which is not the case here. Once again, this is not only expected but needed, since the gradient flow of the Dirichlet energy is a very simple, non-parametric equation (i.e. the heat equation).
> > >
> > > However, the analysis is performed using LFD/HFD characterizations that depend on the Dirichlet energy; as mentioned above, LFD/HFD are simply measures capturing whether the underlying GNN-dynamics is a smoothing (LFD) or sharpening (HFD) process.
> > >
> > > Hope this further clarifies your point.
> > >
> > > Thanks again for your detailed feedback and for engaging with us.

---

### Review · Reviewer_tQzv · 2023-07-06

**Summary Of Contributions:**

This paper explores the expressivity power of GNNs. Specifically, it answers whether simple message-passing GNNs can show a behaviour different from that of oversmoothing, in which the node features start to homogenize as more layers are added to the GNN, irrespectively of the initial feature values. By interpreting time-continuous GNNs are gradient flows of an energy function, the authors show that certain simple GNNs can indeed show oversharpening, an effect contrary to oversmoothing. Moreover, the authors extend their results to more complex GNNs with non-linear activation functions, and empirically validate the theoretical findings, showing that a properly parametrized simple GNN can outperform tailored oversmoothing solutions, casting doubts on commonly-adopted benchmarks and solutions in the literature.

**Audience:**

Yes

**Broader Impact Concerns:**

I don't think there are any broader impact concerns for this work.

**Claims And Evidence:**

Yes

**Requested Changes:**

I think this is good work, and that the results are of interest for the community. In its current state, I wouldn't require any major changes for recommending its acceptance.

However, I do strongly believe that it would be extremely beneficial for the possible outreach of this work to remove extra content that is not fully utilized (like additional unused parametrizations), in pursuit of strengthening the message and focus of the manuscript. As it stands right now, I think it is easily accessible to only a small fraction of the community.

**Strengths And Weaknesses:**

**Strengths**
- The paper's results are relevant for the community, and challenge widespread beliefs by the GNN community.
- The story of the paper is rather intuitive, going from gradient flow for time-continuous GNNs, to discrete-time GNNs (layers) and non-linearities.
- The paper does a good job at providing explanations and intuitions for the technical results.
- Empirical results corroborate the efficacy of the parametrization studied.

**Weaknesses**
-  I find confusing having Q1 and Q2 in the manuscript, as to my eyes they are intimate related and Q2 may imply Q1.
- The paper tries to be really general, yet I feel some parts are not clear, and it would benefit a lot from simplifying the formulations and extending the main explanations of the paper. For example:
  - The formulation in Eq. (1) is quite general, yet all results in the main paper assume a simplified version of it, unless I missed something.
  - Point (i) at the beginning of page 7 is not immediately obvious to me.
  - Non-trivial derivations, e.g. Eq. (6), are not written in the appendix nor main paper.
- I feel that the assumption of the adjacency matrix A being symmetric can be easily missed/forgotten, and more emphasis should be put into it. An easy way to fix it, would be to focus on _undirected_ graphs, whose directed version is a symmetric adjacency matrices.

_Disclaimer:_ I have not checked the correctness of the proofs in the appendix.

---

> ### Author Response · Authors · 2023-07-07
> **Thanks for the review, and response part I**
>
> Thanks for the very detailed review and feedback. We have provided a list of the general updates to the manuscript above, while here we address all your points in order.
>
> Concerning weaknesses:
>
>
> “_I find confusing having Q1 and Q2 in the manuscript_ "
>
> Q.1 and Q.2 are indeed related, and in a broader sense, one could claim that a positive answer to Q.1 would also imply a positive answer to Q.2. Nonetheless, we still believe that it may be beneficial to keep the questions separate.
>
> Q.1 pertains to the frequency response of a GNN and becomes relevant in the context of heterophilic graphs, while Q.2 pertains to possible asymptotic behaviours beyond over-smoothing. Namely, if for a GNN the features _at some layer (time) t_ have (normalized) Dirichlet energy higher than the Dirichlet energy of the input features, then we can say that the GNN is capable of enhancing the high frequencies. However, the asymptotic behaviour (i.e. limit of infinite layers) may still be undetermined.
>
> To provide some further clarification _we have added a sentence at the beginning of pag. 2_ to clarify that Q.2 can somehow be thought of an implication of Q.1 but decided to keep them separate to emphasize the differences and the connections to heterophily and over-smoothing, respectively.
>
> “_The formulation in Eq. (1) _is quite general, yet all results in the main paper assume a simplified version of it_ ”
>
> Thanks for this comment. This is an important point that we have addressed in the revised version. First, Eq. (1) is used in its generality on two occasions: (i) proving when its linear version is the gradient flow of an energy, and (ii) deriving Theorem 6.1, which pertains to the analysis of the monotonicity of non-linear layers. The simplified versions of Eq. 1 are instead used for the derivations of the exact LFD/HFD behaviours for both the discrete and continuous cases (Theorem 4.1 and Theorem 5.1).
>
> We think that starting from a general equation is helpful, since it highlights when this equation can be casted as gradient flow, and we further prove that even in the non-linear case we can derive some monotonicity properties. However, we fully agree that some further clarification and separation could help the presentation. In this regard, **we have modified the manuscript as follows** (we copy here the relevant part in the general comments above):
>
> - At page 7 above the statement of Theorem 4.3, we have added a paragraph “Similarly to..” where we clarify that the exact spectral analysis in the time continuous and time discrete cases is performed for a simplified version of the gradient flow system. We have also stated more visibly the assumption used throughout the rest of Section 4 and Section 5 to help the reader. We note that the results can be extended to versions of the more general gradient flow equations of Proposition 1 quite trivially (e.g. when
>  and $\boldsymbol{\Omega}, \mathbf{W}$ commute), however this could deviate too much from our story. In fact, the purpose of our work is showing that there exist simple graph convolutions that can enhance the high frequencies and induce behaviours other than over-smoothing. On the other hand, we highlight that the results in Section 6 for the non-linear layers hold in the generality of Eq. 1.
>
> - In line with the previous point, we have modified Eq. (10) at the beginning of Section 5 to be the simplified gradient flow system we consider for Theorem 5.1.
>
>
> “ _Point (i) at the beginning of page 7 is not immediately obvious to me._ ”
>
> We agree with the reviewer that point (i) may not be obvious before analysing the equations. We have then removed item (i) from pag. 7 and instead added a comment at pag. 8 below eq. 11, starting “In fact we note..”. In light of Eq. 11 it should now be more transparent that all projections of the features into the kernel of $\mathbf{W}$ stay invariant since the zero eigenvalue of $\mathbf{W}$ kills the dependence on time (layers).
>
> “ _Non-trivial derivations, e.g. Eq. (6), are not written in the appendix nor main paper._ ”
>
> Thanks for the feedback, we have now added a derivation in Appendix B.1 and a relevant pointer in the main text.
>
> " _I feel that the assumption of the adjacency matrix A being symmetric can be easily missed/forgotten_ (...) "
>
> Thanks for the feedback. We have already specified in Section 2 that graphs are always taken to be undirected. Besides that, we have now added an explicit assumption above Theorem 4.3 pag. 7 to specify that the results in Theorem 4.3 and Theorem 5.1 are taken when A is the symmetrically normalized adjacency matrix.

---

> ### Author Response · Authors · 2023-07-07
> **Response part II**
>
> “ _However, I do strongly believe that it would be extremely beneficial for the possible outreach of this work to remove extra content that is not fully utilized_ (...) "
>
> Thanks for the comments. We have revised the manuscript, as detailed in the general response above and in the rebuttal, to make more transparent when we are considering general gradient flow equations, and when instead we are looking at GCN-type of gradient flows to derive exact spectral bounds.
>
> In particular, we highlight that Section 5 now directly studies the simple class of GCN-like gradient flows that are adopted for the main Theorems 5.1 and 5.3. Similarly, we have added a paragraph and an assumption before Theorem 4.3 to further stress that the exact asymptotic analysis is derived for this subset of gradient flow equations.
>
> On the other hand, we again note that the extra parameterizations are used for both deriving Proposition 4.1 and the monotonicity of the energy in the case of non-linear layers (Theorem 6.1).
>
> We hope that our rebuttal and revisions have addressed your valuable concerns and we are happy to engage further in discussions about any standing point.

---

> > ### Comment · Reviewer_tQzv · 2023-07-11
> > **Thanks for the response**
> >
> > Dear authors, thanks a lot for the prompt reply.
> >
> > After reading every reply and checking out the new manuscript, I consider all my concerns (which mainly involved the accessibility of the content to broader audiences) rather solved, and those unsolved details come down to my own personal taste.
> >
> > I would consider necessary clarifying the technical concerns described by reviewer fVAu during the rebuttal period, but from my side I don't need any further clarifications. Thanks again for the kind reply.

---

> > > ### Author Response · Authors · 2023-07-11
> > > **thanks**
> > >
> > > Thanks again for the review and for engaging with us.
> > >
> > > We believe we have clarified all technical concerns of reviewer fVAu and have already modified the manuscript accordingly by explicitly deriving some points that were asked about, however we of course stand ready to address any further doubt or concern!

---

### Review · Reviewer_azt4 · 2023-07-09

**Summary Of Contributions:**

This paper proposes an analysis and re-evaluation of basic MPNNs such as GCN via an energetic perspective. The paper shows that by setting the appropriate weights in models like GCN, they can be viewed as a gradient flow that is either smoothing (a commonly known observation), but also over-sharpening (a surprising result shown in this paper).

Thus, the authors suggest that despite the common belief that simple MPNNs like GCN are bound to always over-smooth, such issues can be avoided using their proposed parameterization.

The authors present several experiments that show the effectiveness of viewing MPNNs as gradient flows, resulting in improved performance compared to simple, known models, like GCN.

**Audience:**

Yes

**Claims And Evidence:**

Yes

**Requested Changes:**

I have read the whole paper including the appendix. I enjoyed reading it, and I think that addressing my questions in my review will improve the paper.

**Strengths And Weaknesses:**

**Strengths**:
* The paper is highly organized and very well written.
* It seems that the general concept proposed here is novel and can also be applied to other MPNNs.
* The paper is theoretically rigorous, as well as practically meaningful, because of the various experiments that show the benefit of the interpretation of MPNNs as gradient flows and the inspection of their energy.

**Weaknesses / Questions**:
* Right before the end of page 1, the authors say that "Several works have gathered evidence that graph-convolutional models seem to struggle on some heterophilic
graphs". Can you please expand on that point? Is it not true for all or most of heterophilic graphs? Are there examples of heterophilic graphs where basic MPNNs (e.g. GCN) do not struggle?
* With respect to signed message passing architectures, the authors may consider adding a reference to "Magnet: A neural network for directed graphs"
* With respect to repulsive and attractive forces, the authors may consider adding a reference to "ACMP: Allen-Cahn Message Passing for Graph Neural Networks with Particle Phase Transition"
* With respect to learning low/high pass filters, the authors may consider adding references to "Revisiting heterophily for graph neural networks", "Improving Graph Neural Networks with Learnable Propagation Operators"
* The authors call subsection 4.1 "non-parametric" and to 4.2 "parametric". I think that the terms used here may be confusing to some readers because usually the notion of "parametric models" is used for closed-form models (e.g., exp(-theta) where theta is a scalar parameter). Perhaps it would be useful to distinguish between the two?
* In equation 5 the authors add a source term but I am not sure it is used later. Is it a beneficial term? Also, how is the source term here different than the residual connection made in "Simple and Deep Graph Convolutional Networks", and the source term in "GRAND++: Graph Neural Diffusion with A Source Term"?
* If I understand correctly, the authors use the same weights W throughout all time steps. May I ask what is the influence on the theoretical results if W can change at each time step, and how may it influence the empirical results?
* In figure 2, what is cornell_old ? is it just a typo?
* Is it safe to assume there always exists some desired energy of an optimal state of an MPNN? If so, can such energy always be achieved using the proposed method?
* In the appendix, the authors define the "GRAFF" model, but it is not shown later in table 3.

---

> ### Author Response · Authors · 2023-07-09
> **Thanks for the review, response part I**
>
> Thank you for the detailed review. We have already modified the manuscript (the list of changes appear below the abstract). Here we address all your additional questions/comments:
>
> - " _Right before the end of page 1, the authors say that "Several works have gathered evidence that graph-convolutional models seem to struggle on some heterophilic graphs". (...)_ "
>
> There are more recent heterophilic benchmarks that have been introduced, where graph convolutional models are not as bad as one might expect and so one should be careful before jumping to broad statements. To avoid confusion, we have added a reference to this recent paper (Platonov et al) in the last paragraph of page 1.
>
> - " _With respect to signed message passing architectures (...)_ "
>
> Thank you for the pointer, we have added a citation in the related work section.
>
> -  " _With respect to repulsive and attractive forces, the authors (...)_ "
>
> Thank you for the pointer, we have added citations in the related work section.
>
> - " _With respect to learning low/high pass filters, the authors may (...)_ "
>
> Thank you for the pointer, we have added citations in the related work section.
>
> - " _In equation 5 the authors add a source term but I am not sure it is used later (...)_ "
>
> Thank you for the comment. First, we note that the introduction of a source term is mainly motivated by the fact that we wanted to consider a family of energies that also included, as a special case, the energy used in classical label propagation approaches. We also note that we have already established a connection with “Simple and Deep Graph Convolutional Networks” below eq. (1). We highlight that our reasoning for including a source term was to recover classical settings and emphasize how even with a source term one could satisfy the gradient flow constraints (and the associated weight matrices need not to be symmetric).
>
> To improve the manuscript according to other reviewers’ feedback, we have now simplified the presentation in Section 5 and considered only the modified class of GCN-type of gradient flows (no source term) to which the exact spectral analysis of Theorem 5.1 applies. The more general class of gradient flow equations (including a source term) is instead studied in Section 6, where monotonicity properties of the energy have been derived even in the nonlinear case.
>
> - “ _If I understand correctly, the authors use the same weights W throughout all time steps. May I ask what is the influence on the theoretical results if W can change at each time step, and how may it influence the empirical results?_ ”
>
> Thanks for the question. If we have no control/assumptions on the form that the weight matrices take over time, then extending the theoretical results may be non-trivial. From a physical perspective, this is equivalent to assuming that the associated energy now has an external dependence on time that often complicates analytical treatment. This is a direction we have been thinking about as a follow-up of the current work, especially trying to focus on ways to control the time dependence of the weights.
>
> From an empirical perspective, all our results in Table 1 and Table 3 are with weights shared across time steps, while for all the baselines the weights are taken to be different and independent for each layer. The results seem to suggest that, for these tasks and benchmarks, there is no price to be paid in terms of performance when sharing the weights. We have also conducted some experiments with the same gradient flow model and allowing the weights to change at each time step (layer) and we have registered no improvement.
>
> - " _In figure 2, what is cornell_old ? is it just a typo?_ "
>
> This is just to point out that the dataset used is the "old" one – a modification to the dataset was performed a few months ago, however all baselines reported in this paper based their results on the older format.

---

> > ### Author Response · Authors · 2023-07-09
> > **Response part II**
> >
> > - " _Is it safe to assume there always exists some desired energy of an optimal state of an MPNN? If so, can such energy always be achieved using the proposed method?_ ”
> >
> > This is an interesting question, and it sort of nails down the whole motivation for this paper. In this paper we have shown how to interpret graph-convolutional models through the lenses of gradient flow of simple energies and have validated that these models might actually suffice to provide optimal states for most node classification tasks. While the emphasis was on proving how and when existing configurations can be interpreted as gradient flow of a parametric energy, the other direction of building entirely new models as gradient flows of some energy is the one that needs more exploration.
> >
> > Certainly there might be tasks where the class of energies we have introduced in this paper could not lead to the desired solution. For example, the family of energies we have studied here amount to quadratic (bilinear) terms leading to gradient flow equations that contain linear terms. It is possible to generalize this construction to involve more non-linear terms at the level of the energy whose associated gradient flow equations lead to the optimal state/classifier for problems other than node classification for example.
> >
> > On a more general level, if we base our intuition on physics, it is probably safe to assume that most problems would admit a gradient-flow description with an optimal state/classifier being the minimizer of some (more or less complicated) energy. The tradeoff one has to figure out, and we hope that our paper can be a first step in building this theory/framework, is how to parameterise the class of energies in a general yet efficient form that can ultimately improve the shape of the loss and hence simplify the learning problem. This is a very broad direction we wish to investigate in the future.
> >
> > “ _In the appendix, the authors define the "GRAFF" model, but it is not shown later in table 3._ ”
> >
> > Thanks for the comment, we have changed Table 3 accordingly and added a paragraph below eq. (35) in the appendix to highlight the differences.
> >
> >
> > Thank you for the detailed feedback and we are happy to engage in discussion to address any standing doubt.

---

> > ### Comment · Reviewer_azt4 · 2023-07-09
> > **Thank you for your response**
> >
> > Dear authors,
> > I thank you for your fast and detailed response.
> >
> > Thank you for the clarification regarding heterophilic datasets.
> >
> > Can you please clarify what are the changes to the cornell dataset? (and possibly others if I understand correctly)
> > If the changes are specific to your paper, it would be great if you can specify them. If the data source has changed, can you please provide a short description of the modification?

---

> > > ### Author Response · Authors · 2023-07-11
> > > **Clarification on Cornell dataset changes**
> > >
> > > This was due to an update of the graph adjacency file for only the Cornell dataset provided by Pei et al 2020. See this commit on 17th July 2022 - https://github.com/graphdml-uiuc-jlu/geom-gcn/commit/3cc57103d2b71f799bb74f34e200ec0f26363269. All baselines in our experiments used the earlier dataset so to maintain the correct comparison we use the earlier version. To remove any confusion we will; update the chart to remove "_old", add a clarifying comment to Appendix E.1 and ensure the data loaders in our released code catch the correct files.

---

> > > > ### Comment · Reviewer_azt4 · 2023-07-11
> > > > **Thank you**
> > > >
> > > > Dear authors,
> > > >
> > > > Thank you for the important clarification.
> > > >
> > > > I have no further standing questions.

---

### Decision · Action_Editors · 2023-08-05

**Recommendation:** Accept as is

**Comment:**

All 3 reviewers agree that the claims are supported by clear evidence, that the work is relevant to the TMLR community, and recommended to accept the manuscript. All the concerns and clarifications asked by the reviewers were addressed during the rebuttal and revisions. I thus recommend to accept the paper as is.

**Audience:**

Yes.

**Claims And Evidence:**

Yes.